# The Influence of Polyphenols on Atherosclerosis Development

**DOI:** 10.3390/ijms24087146

**Published:** 2023-04-12

**Authors:** Agnieszka Ziółkiewicz, Kamila Kasprzak-Drozd, Robert Rusinek, Ewa Markut-Miotła, Anna Oniszczuk

**Affiliations:** 1Department of Inorganic Chemistry, Medical University of Lublin, Dr Wiotolda Chodźki 4a, 20-093 Lublin, Poland; 2Institute of Agrophysics, Polish Academy of Sciences, Doświadczalna 4, 20-290 Lublin, Poland; 3Department of Lung Diseases and Children Rheumatology, Medical University of Lublin, Prof. Antoniego Gębali 6, 20-093 Lublin, Poland

**Keywords:** atherosclerosis, cardiovascular diseases, polyphenols, inflammatory diseases, oxidative stress, gut microbiota

## Abstract

Polyphenols have attracted tremendous attention due to their pro-health properties, including their antioxidant, anti-inflammatory, antibacterial and neuroprotective activities. Atherosclerosis is a vascular disorder underlying several CVDs. One of the main risk factors causing atherosclerosis is the type and quality of food consumed. Therefore, polyphenols represent promising agents in the prevention and treatment of atherosclerosis, as demonstrated by in vitro, animal, preclinical and clinical studies. However, most polyphenols cannot be absorbed directly by the small intestine. Gut microbiota play a crucial role in converting dietary polyphenols into absorbable bioactive substances. An increasing understanding of the field has confirmed that specific GM taxa strains mediate the gut microbiota–atherosclerosis axis. The present study explores the anti-atherosclerotic properties and associated underlying mechanisms of polyphenols. Moreover, it provides a basis for better understanding the relationship between dietary polyphenols, gut microbiota, and cardiovascular benefits.

## 1. Introduction

A process mainly observed in developed or developing countries [1], atherosclerosis is considered a major cause of morbidity. Population growth, an aging population and epidemiological changes are increasing the number of deaths from cardiovascular disease (CVD) [2]. Atherosclerosis is a chronic inflammatory disease characterised by damage and hardening of the inner layer of the arterial wall and the accumulation of plaques, which can result in thrombosis. The multifactorial disease is primarily due to lipid accumulation and the development of chronic inflammation in the vessels [1].

The pathogenesis of atherosclerosis is related to the migration of monocytes, their transformation into macrophages and endothelial activation. The accumulation of oxidised low-density lipids, becoming foam cells, results in the formation of atherosclerotic plaque and obstruction of normal blood flow [3]. Activation of macrophages and smooth muscle cells leads to a release of hydrolytic enzymes, cytokines, chemokines and growth factors that induce focal apoptosis [4]. The atherosclerotic process is accompanied by an immune response to low-density lipoprotein LDL and other antigens, alleviating or exacerbating the course of the disease [5]. The resulting atherosclerotic plaques are divided into those with a predominantly lipid component or fibrous tissue [4]. The aforementioned processes lead to cardiovascular disease [3]. 

With lifestyle changes that include smoking cessation, increased daily physical activity, and reduced consumption of fatty foods, there has been a decline in cardiovascular disease CVD mortality in the United States in the 21st century. Despite this, it is estimated that almost 50% of the adult population will be burdened with some form of CVD by 2023 [6]. Many studies have been conducted to date showing the impact of quality of life and diet on the incidence of various chronic diseases, including diabetes, hypertension and cardiovascular disease. According to the 2015–2020 guidelines, the recommended daily intake of fruit is two cups, yet Americans continue to consume more than the recommended amount of meat compared to fruit and vegetables. 

The different phases in the pathogenesis of atherosclerosis are all promising therapeutic targets. Statins that lower LDL cholesterol and have additional pleiotropic effects are widely used. However, there is a considerable risk in patients on statin therapy, with some individuals unable to achieve target LDL cholesterol goals or cannot tolerate the medicine. Moreover, high statin dose may cause side effects such as rhabdomyolysis, non-allergic rhinitis and hyperglycemia [7]. Clinical trials using orally active HDL-raising agents was disappointing, just like trials on inhibitors of cholesteryl ester transfer protein [8]. Due to atherosclerosis being an inflammatory disorder, approaches that dampen inflammation are also being pursued [9]. However, such approaches will have to be restricted to high-risk patients, because of the risks associated with manipulating systemic inflammation. For this reason, scientists are constantly looking for alternative ways to prevent atherosclerosis. A link between antioxidant consumption and the inhibition of atherosclerosis was recently discovered. Accordingly, nutrient-containing polyphenols, especially anthocyanins and flavonoids, have potential in countering cardiovascular disease. Furthermore, phenylalanines, consisting of aromatic carbon and hydroxyl rings, are known to have an important function in scavenging reactive oxygen species. Organic-rich fruits, vegetables, teas or traditional medicinal herbs should therefore form an indispensable part of everybody’s diet [10]. Of note, extra virgin olive oil is particularly rich in polyphenols and has shown strong antioxidant properties. 

A poor diet causes many negative effects, contributing to the development of many diseases. Numerous studies have shown the positive effects of the Mediterranean diet on cardiovascular health. It has been found to reduce plasma lipid, glucose and blood pressure concentrations, as well as the plasma concentrations of inflammatory biomarkers (interleukin IL-6, vascular cell adhesion protein 1(VCAM 1) and Intercellular Adhesion Molecule 1 (ICAM-1). The Mediterranean diet is characterised by a high content of cereals, vegetables (rich in fibre), fruits, and seafood (sources of omega-3 polyunsaturated fatty acids). Consumers of this diet limit red wine, meat, dairy, eggs and sweets. Until the 1960s, the Mediterranean style of eating resulted in the lowest rates of chronic disease and the highest life expectancy in the region [4]. 

In recent years, there has been a surge of interest in the interaction of polyphenols and gut microbiota. They have become the subject of numerous scientific studies due to their potential beneficial effects on the body’s health, in particular, the prevention of cancer and cardiovascular disease [11]. Polyphenols act selectively on microorganisms, inhibiting the growth of pathogens and showing prebiotic effects. The colonic microbiome converts polyphenols into bioactive compounds, resulting in better absorption of them [12]. A relationship between the microbiome and atherosclerosis is observed in the presence of inflammation occurring with intestinal dysbiosis. There is then an increased permeability of the mucosal barrier, thus increasing permeability to bacteria [13]. 

The aim of this review was to summarise current reports on the effects of polyphenols contained in food on the formation, prevention and treatment of atherosclerosis, and to analyze the anti-inflammatory effects of these organic compounds on the aforementioned pathology.

## 2. Atherosclerosis: Risk Factors and the Mechanism of Development

Atherosclerosis is a global health problem. In recent years, there has been a sharp increase in the incidence and consequent emergence of cardiovascular disease. Its causes are mainly related to lifestyle (diet, smoking, physical activity), while developmental factors include modifiable (lipid disorders, diabetes, hypertension, obesity, metabolic syndrome and inflammatory markers) and those beyond human control (age, gender, family history and genetic markers). All of the aforementioned factors affect the function and structure of the glycocalyx (found in the vascular endothelium) and the production of nitric oxide (NO) [13,14]. A diet rich in waxy and high-fat products promotes an increase in total daily caloric intake, which contributes to the development of the disease, while neglecting to add fish, fruit, vegetables or nuts to daily meals also promotes the risk of affliction [15]. In addition to diet, the development of atherosclerosis is influenced by body composition, including the amount of visceral fat, which leads to the development of abdominal obesity [13]. 

A contributing factor to increased body weight is physical inactivity. A sedentary lifestyle worsens the physical condition of the body and increases mortality. The lack of daily sport during the day affects changes in muscle and white adipose tissue, promoting an inflammatory and atherosclerotic environment [13]. The prevalence of obesity in developed and developing countries has shown an increasing trend over the years. There is a proven link between its presence and diseases such as hypertension, diabetes or dyslipidemia. 

Obesity is determined by clinical measurements of the abdomen to waist circumference, giving a normal result of 0.9 in men and 0.85 in women. The pathogenesis of the development of obesity is primarily related to increased daily caloric intake and, in particular, an excess of free fatty acids circulating in the blood, leading to chronic inflammation [16,17]. Developing adipose tissue releases tumour necrosis factor, interleukin 6 (IL-6), resistin, leptin, vascular endothelial growth factor, vascular endothelial growth factor (VEGF) and free fatty acids, as well as adipokines, which stimulate the formation of insulin resistance and endothelial dysfunction. Its cells are mainly responsible for processes of inflammatory-anti-inflammatory balance and oxidation and antioxidation reactions, so their damage is associated with impaired vasoconstriction and fibrinolysis. Factors that mainly contribute to cell degradation include elevated LDL cholesterol, tobacco smoke, obesity, hypertension, hyperglycaemia and oxidative stress [18,19]. The stated factors also cause an increase in the inflammatory marker C Reactive Protein (CRP), which further increases the risk of peripheral vascular disease and diabetes [16]. 

Tobacco smoke is one of the most important risk factors for atherosclerosis. Numerous studies indicate that even passive smoking can lead to myocardial infarction as a consequence. This is associated with increased thrombocyte activity and enhanced inflammation in the body [16]. The chemical components of smoke have oxidative and inflammatory properties that directly cause endothelial damage. In addition, they promote the onset of oxidative stress by inducing Cyclooxygenase-2 (COX-2) expression, affecting the activity of enzymes such as endothelial Not Otherwise Specified (eNOS), thereby resulting in irreversible protein modification and alteration of intracellular pathways. Tobacco smoke damages the vascular endothelium by affecting membrane permeability, resulting in increased LDL levels that promote the deposition of atherosclerotic plaques [3,14].

Diabetes poses a major threat to human health and life. Chronic hyperglycaemia leads to metabolic disturbances of all the body’s macronutrients, resulting in the multistage development of atherosclerosis and associated cardiovascular diseases [1]. Studies have shown damage to the endothelial glycocalyx in patients diagnosed with type 1 diabetes, leading to increased vascular permeability to LDL [14]. A link between the presence of insulin resistance and atherosclerosis has been shown in dyslipidaemic patients. Low levels of small LDL are a scientifically proven risk factor. The atherogenic modification of these particles causes lipid accumulation in the vessels. This alteration leads to the interaction of LDL with proteoglycans, with a consequent reduction in affinity for the correct receptor [1]. 

The oxidative stress created in the body by damaging endothelial cells initiates the release of many molecules, including chemokines, which attract monocytes to the damaged area of the vessel. By interacting with adhesion molecules present in the blood, they cause adhesion to the endothelium. The mature macrophage absorbs previously accumulated abnormally phagocytosed LDL cholesterol, causing foam cells to form and, consequently, atherosclerosis [18]. Advanced glycation end products (AGEs) are produced in the presence of high blood glucose values, oxidative stress or an inflammatory response. These compounds are difficult to metabolise and accumulate, accelerate the migration of monocytes and stimulate macrophages to produce cytokines, accelerating the progression of atherosclerosis. AGEs promote an increase in endothelin-1, increasing vasoconstriction. Moreover, they reduce NO levels, decreasing vasodilation. Studies show that AGEs are closely linked to the pathophysiology of atherosclerotic lesions. They induce the production of VEGF, initiating pathological neovascularisation and tissue factors and activating the coagulation system. Microvascular complications, such as renal and retinal manifestations, pose an additional risk of uncontrolled glycaemic surges [1,18].

Hypertension is one of the most important factors in the development of cardiovascular disease, causing morphological changes in the arterial endothelium, as well as hypertrophy of the smooth muscle layer in the middle sheath. Increased pressure also generates a change in membrane permeability, leading to increased LDL entry [14,20]. Endothelial dysfunction is further caused by a decrease in nitrogen oxide levels. Its deficiency negates thrombocyte reduction, activates thrombogenic factors and disrupts metabolic homeostasis by increasing triglycerides levels and decreasing insulin synthesis [19,21].

Magnesium (Mg) has a regulatory function in blood pressure. Alteration of endogenous magnesium results in modification of vascular tone and changes in arterial pressure. Relatively small changes in intracellular, as well as extracellular ion concentration, significantly affect vascular tone, contractility, reactivity and growth. All of the above-mentioned changes, which have a negative effect on arterial pressure, cause it to rise. Hypomagnesaemia impairs the production of prostaglandin E1, which has a role in vasodilation and thrombocyte aggregation [20]. In addition, low Mg concentration impairs endothelial vasodilatory function and calcium ion influx and accumulation, leading to vasoconstriction and increased vascular resistance. Hypomagnesaemia leads to the development of insulin resistance and hyperglycaemia. In addition, it exacerbates changes in lipid metabolism and enhances the transport of lipoproteins (LDL) through the endothelial layer so that they accumulate in the vascular lumen, causing arterial stiffness and the development of atherosclerotic lesions [20]. Hypertension, hyperlipidaemia, diabetes and tobacco smoke increase oxidative stress. Through the synthesis of pro-atherogenic cytokines (tumour necrosis factor-alpha (TNF-alpha), IL-1, IL-6), adhesion molecules (VCAM-I and ICAM-I) and chemokines (monocyte chemotactic protein-2 (MCP-1), NO production is inhibited, resulting in vasoconstriction [3].

Current research shows that men are more than twice as likely to have HDL lipoproteins and overall higher LDL levels across all age categories. This is associated with an enhanced risk of developing hypertension. Non-invasive imaging methods (ultrasound) reveal higher plaque thickness in men. The reduced risk of atherosclerotic plaques in women is influenced by female sex hormones (mainly oestrogen). Studies have shown that ovarian excision significantly increases plaque sise. In addition, oestrogen decreases CD16+ monocytes expression. Men compared to women therefore have more circulating CD14+ monocytes and CD16+ monocytes in the blood, which are associated with endothelial dysfunction. All of the aforementioned anomalies result in a higher and earlier risk of atherosclerosis in the male sex [22]. 

### 2.1. Atherosclerosis as an Inflammatory Disease

Atherosclerosis is an inflammatory chronic vascular disease primarily associated with oxidative stress and endothelial dysfunction that leads to manufacturing CVD [23,24]. Numerous studies confirm that hyperlipidaemia alone is not sufficient for atherosclerosis development; rather, inflammation becomes an additional stimulus [25]. It is mainly triggered by the interaction of the immune system with the cardiovascular system and local vascular lesions [26]. 

The atherosclerotic process is characterised by the accumulation of macrophages, smooth muscle cells and lymphocytes in the vessel wall. The high number of inflammatory cells results in excessive production of reactive oxygen species and cytokines. Current research shows that the development of atherosclerotic plaque is influenced not only by the deposition of lipid, but also by the activation of the immune and inflammatory response. The chemokine receptors CCR2, CCR5 and CX3CR1 and their ligands recruit inflammatory monocytes into atherosclerotic plaques. As a result of ischaemic changes, the bone marrow and spleen produce pro-inflammatory monocytes in increased numbers, which further exacerbate the atherosclerotic process. In addition, circulating monocytes in the blood become hyper-reactive [24]. 

Macrophages play a major role in atherosclerotic plaque development. They are divided into two classes (M1 and M2), with the former showing increased lipid accumulation. Macrophages are stimulated by lipopolysaccharide or gamma interferon and then polarise to the M1 phenotype via the nuclear factor B or signal transducer and activator of transcription 1 pathway. The M1 class increases glucose uptake, impairing oxidative phosphorylation and releasing reactive oxygen species. M2 macrophages secrete anti-inflammatory cytokines (IL-4, IL-10, IL-1). TGF-β secreted by the macrophage inhibits inflammation by decreasing iNOS-specific activity and decreasing iNOS protein production. Studies also confirm the secretion of arginase, epidermal growth factor (EGF), VEGF, IL-6, TNF and IL-1 [27].

The adaptive immune system consists of T cells, B cells and antigen-presenting cells. B cells are divided into type 1 and type 2. Type B1 produces IgM antibodies that protect against atherosclerosis, while type B2 produces IgG antibodies that stimulate the inflammatory process. B cells also have a regulatory function, producing IL-10 [27]. Mast cells contribute to inflammation by producing cytokines and infiltrating leukocytes, increasing plaque surface area as they degranulate. Dendritic cells (DCs) capture lipids and are involved in foam cell formation. Mature DCs activate T cells by presenting antigens and removing apoptotic cells [24,26]. T cells participate in the cellular immune response by responding to antigens presented by APCs and by secreting signaling molecules [24]. Th 1 lymphocytes, termed ‘helper lymphocytes’, play an important role in cellular immunity by producing TNF- α and IFN-γ anti-atherogenic factors [26]. The CD8+ Treg lymphocyte restricts the growth of Th1 lymphocytes and macrophages, hence, demonstrating a protective function [27]. The NLRP3 inflammasome has also been shown to play an important role in the development of inflammation [25]. It is a mutimeric cytosolic protein complex, accumulating molecular patterns associated with damage, which is stimulated by an extracellular trap derived from neutrophils or oxidised lipids and then captured by oxidised LDL (oxLDL) via CD36. It processes the cytokines IL-1β and IL-18 and secretes IL-1β36. Studies have shown NLRP3 expression in platelets and peripheral blood mononuclear cells in patients with atherosclerosis. In addition, activation of the inflammasome is influenced by oxidative stress, lysosome rupture, mitochondrial dysfunction and endoplasmic reticulum stress [25,26]. 

OxLDL binds to antibodies forming immune complexes so that an inflammatory response is induced in macrophages and dendritic cells. As a result, cytokines are secreted, and foam cells are formed. These cells are differentiated from monocytes in the inner vessel wall by the action of chronic lymphocytic leukemia (CCL2) and T-cell chemoattractants. Increased ROS production leads to the infiltration of oxLDL into the endothelium. This interferes with normal endothelial function and monocyte adhesion and infiltration [3].

Toll-like receptors (TRLs) play a highly important role in the innate and adaptive immune response. These are initiated by saturated fatty acids, most notably Apo CIII, which induce signals through TRL2. In addition, modified LDL activates TRL4 in macrophages through a MyD88-dependent or non-MyD88-dependent pathway, secreting the proinflammatory cytokines IL-1β, IL-6 and TNF-α in an NF-kB-dependent manner [24,27]. TLR signaling promotes the production of antimicrobial peptides, proinflammatory cytokines, adhesion molecules, and reactive nitrogen and oxygen species. The dysregulation of TLRs is a key mechanism in the development of inflammation and atherosclerotic processes. Studies show that TLR2 and TLR4 expression is increased in peripheral blood mononuclear cells, monocytes, endothelial cells and vascular smooth muscle cells [26,28].

Proprethrin convertase subtilisin type 9 (PSK9) is an essential element involved in the pathophysiology of atherosclerosis. It is involved in the breakdown of receptors for LDL and increases cholesterol levels. Its expression is regulated by sterol response element binding proteins (SREBP) 1 and 2, as well as peroxisome proliferator-activated receptor (PPAR) α and γ and sirtuin (SIRT) 1 and 6. PCK9 degrades hepatic LDLR and regulates its levels in immune and vascular cells. Additionally, it inhibits foam cell formation and downregulates LDLR expression on the surface of macrophages. Through interactions with platelet CD36, it increases thrombocyte activation and thrombosis. It also acts as a pro-inflammatory mediator by enhancing TLR4 and NF-κB expression [26].

The resolution of inflammation in atherosclerosis is a multistep and multicomponent process. A number of endogenous factors and protein mediators (annexin A18. IL-10), carbon monoxide and hydrogen sulphide are needed to reduce inflammation. SPMs are synthesised at the onset of acute inflammation. The balance between SPMs and leukotrienes, and prostaglandins determines how long inflammation is maintained. They are created from arachidonic acid (AA), eicosapentaenoic acid (EPA), docosahexaenoic acid (DHA) or n-3 docosapentaenoic acid through the action of lipoxygenases (LOX) and cyclooxygenases (COX). Inflammation is exacerbated by the deletion of the key receptors ALX/FPR2, GPR18 or LGR6 [12]. The mechanism of the atherosclerotic process is presented in Figure 1.

### 2.2. The Impact of Oxidative Stress on the Development of Atherosclerosis

Oxidative stress occurring in the human body leads to the development of many serious diseases, such as lung disease, cancer and atherosclerosis. It involves a series of reactions leading to the formation of reactive oxygen species. One mechanism of oxidative stress is the production of molecules that directly damage structural proteins, membrane lipids, nucleic acids and enzymes. These are the superoxide anion radical (O_2_^−^), hydroxyl radical (OH) and hydrogen peroxide (H_2_O_2_). The second is the modulation of the activity of many proteins and abnormal redox signaling, in which cell-generated oxidants act as secondary messengers, and changes in cell phenotype are induced [1,10]. An imbalance between oxidant and antioxidant systems leads to the formation of more reactive oxygen and nitrogen species (RONS) [2]. Potential sources of reactive oxygen species include the respiratory chain, NADPH oxidase, xanthine oxidase, endothelium-associated NO synthase (eNOS), cyclooxygenase, myeloperoxidase and lipoxygenase [10].

The vascular wall is equipped with oxidative systems, including xanthine oxidase and NADPH (Nox). The former acts as an electron acceptor, forming O_2_^−^ and H_2_O_2_ and producing uric acid, which is essential for foam cell formation. The second system reduces O_2_ to superoxide anion, providing the main source of RONS in the vessel wall. Defects in the function of mitochondrial respiratory chain enzymes increase the production of reactive oxygen species, which in turn induce the expression of the inflammatory factors MCP-1, ICAM-1 and IL-1, producing ROS, thus creating a closed causal circle [2,3]. The production of reactive oxygen species is dependent on NADPH oxidase. Early in the atherosclerotic process, they are a major cause of endothelial dysfunction and vascular wall remodeling [10,13]. 

Beyond the cell wall, ROS also act on elements deep inside the cell. DNA is an extremely important part needed for the functioning of every cell. However, it is highly susceptible to the negative effects of ROS. Single- and double-strand breaks cause tremendous damage in cells, which contributes to increased chromosome damage in peripheral lymphocytes. The presence of mitochondrial DNA in the matrix and the lack of protective histones greatly increase the potential for the degenerative effects of mtROS [17].

Nitrogen compounds also have a negative effect on the cell. In the synthesis of this compound oxide, the cofactor tetrahydrobiopterin (BH4) is essential. In oxidative stress, there is an increase in the amount of RFTs produced, which, by oxidizing the BH4 cofactor, causes NO degradation. The presence of its oxidised form reduces the bioavailability of nitric oxide, playing an important role in the pathogenesis of atherosclerosis. Numerous studies suggest that cellular deprivation of NOX 1 and 2 results in the production of reactive oxygen species. NADPH oxidase isoforms affect vasodilatory function in the vasculature and alter nitric oxide availability. Increased NOX expression results in an increase in p22phox accompanied by an increase in reactive oxygen species production, which modulates atherosclerotic plaque stability and the presence of oxLDL [5]. Induction of oxLDL by ROS allows passage through the damaged endothelium into the vasculature and enables the differentiation of monocytes into macrophages. These, in turn, by means of scavenger receptors absorbing oxLDL in an uncontrolled manner, become foam cells capable of accumulating into atherosclerotic plaques [10,29] (Figure 2). In the presence of T lymphocytes, oxLDLs activate the immune response. In addition to exhibiting proatherogenic properties, they stimulate leukocyte adhesion, promote vasoconstriction, enhance platelet aggregation and increase the expression of growth factors, thus contributing to vessel wall remodeling [10]. Mitochondrial lipids become the main target of RFT damage. Cardiolipin, due to the presence of a large amount of unsaturated fatty acids, becomes very vulnerable to external agents. The high content of its oxidised form is a pro-inflammatory signal, activating the expression of leukotrienes and adhesion molecules (ICAM-1 and VCAM-1) [17].

Mitogen-activated protein kinases (MAPKs) are enzymes belonging to a group of redox-sensitive signaling molecules involved in the transmission of signals related to various processes relevant to the pathogenesis of atherosclerosis. MAPKs are kinases that regulate gene transcription, protein biosynthesis, cell division, differentiation and cell survival and apoptosis. Studies show that the intracellular reactive oxygen species produced cause their activation, thereby leading to degenerative processes in cells. MAP kinase-dependent stress-sensitive transcription factors are AP-1, STAT-3 and NF-κB. Their activation leads to the transcription of genes involved in inflammatory response mechanisms [10]. ROS primarily induce the synthesis of the pro-inflammatory cytokine TNF- α, which contributes to the activation of NF-κB [13]. The NF-κB pathway is also involved in the inflammatory response when extracellular signal-regulated kinase-5 (Erk-5) activity is reduced. Reduced kinase activity, in turn, leads to increased RFT formation in endothelial cells. The endothelium then becomes a source and target of ROS attack [14,17]. The susceptibility of endothelial cells to NO through the presence of eNOS results in enhanced expression of granular membrane protein-140, so that neutrophils are more readily bound to the cell surface. What is more, the induction of NFκB by oxidative stress results in the expression of adhesion molecules, which then induce the production of mtROS. In addition, iNOS expression and NADPH oxidase activity increase [17].

AGEs (end products of advanced glycation) represent an important abnormality in the body, leading to the formation of reactive oxygen species and driving the progression of arteriosclerosis. The pathophysiological changes occurring in the vasculature are associated with progressive degenerative processes that originate in diabetes. High uncontrolled glycaemic levels lead to the development of diabetic microvascular disease. The most prominent components are the products of non-enzymatic glycation of proteins and lipids and the deposition of AGEs in the wall. Recent ongoing research has shown that the development of atherosclerosis is influenced by an isolated AGE-‘RAGE’ surface receptor. It acts as a signal-transduction receptor for inflammatory molecules. The ligand–RAGE interaction has the effect of modulating vascular properties, and the consequence for such a connection becomes cellular activation, which leads to the induction of oxidative stress [15]. AGEs have been shown to alter the properties of major matrix proteins, collagen, vitronectin and laminin, through intermolecular AGE-AGE cross-links [16]. 

In these pathological changes, the miRs play an extremely important role as they are sensitive to oxidative stress, and studies have shown elevated miR-92a in atherosclerotic plaques. This, in addition, initiates endothelial cell activation and inflammation by increasing the expression of ICAM-1, MCP-1 and NF-κB. In contrast, miR-155 down-regulates NO levels in ECs and induces apoptosis by activating caspase-3 [14]. Studies have revealed the inhibitory effects of miR-19b-3p, miR-221-3p and miR-222-3p on proliferator-activated coactivator gamma protein expression. In addition, miR-34a and miR-383 increase oxidative stress by repressing sirtuin, while preventing PGC-1α deacetylation. The effects of advanced glycation end products and ox-LDL on miRs are noteworthy. They induce miR-92a, silencing HO-1. Moreover, ox-LDL acts on miR-155 and IncRNA, which interact with Opa protein five antisense RNA 1 to accelerate endothelial-damaging responses and inflammation in the TLR4/NFκB signaling pathway. Numerous studies have shown the negative effects of miR-22, miR-23a, miR-26a and miR-34a on cells. This is because they increase mitochondrial ROS and induce apoptosis. Inflammation and oxidative stress upregulate miR levels in cells [17].

## 3. Classes of Phenolic Compounds and Their Role in Human Health

### 3.1. Structure and Classification of Phenolic Compounds

Polyphenolic compounds are a broad group of organic compounds that are secondary plant metabolites. Formerly considered to be anti-nutritional substances, they are now of increasing interest mainly due to their considerable and diverse bioactivity [30]. In plants, they perform a variety of functions, including supporting protection against ultraviolet radiation, excessive drought and too-low temperatures. Polyphenolic compounds are also involved in the defense against pathogens, parasites and herbivores [31]. Until now, more than 8000 polyphenol structures have been identified [32]. The structure of polyphenolic compounds can be both in the form of simple molecules (for example, phenolic acids, but also highly polymerised molecules (condensed tannins) [31]. A characteristic of the structural structure of polyphenolic compounds is one or more aromatic rings and from one to several hydroxyl groups, giving them an acidic character [30]. The reduced electron density on the oxygen atom of the phenolic group causes the binding energy of hydrogen is much lower than in the case of the hydroxyl group occurring in aliphatic compounds. Phenolic compounds readily donate hydrogen and melt into semiquinones, which are polymerised to color compounds. Due to their ability to transfer protons and electrons, these compounds not only oxidise themselves but also, through the quinones produced by oxidation, can mediate the oxidation of compounds that do not react directly with oxygen. The effectiveness of polyphenolic compounds depends largely on the molecular weight and structure, and concentration [33]. Polyphenolic compounds are formed from primary metabolites by the transformation of shikimate/phenyl propanoic or polyketide pathways [31].

Polyphenolic compounds are a wide and heterogeneous collection of phytochemically active compounds. Depending on the number of phenolic rings and how they are combined, major groups can be distinguished, such as flavonoids (flavonols, flavanols, flavones, flavanones, isoflavones, anthocyanins) and non-flavonoids (phenolic acids, lignans, tannins, stilbenes, xanthones) [34]. The largest group of compounds belonging to polyphenolic compounds are flavonoids [35]. About 6000 different flavonoid structures have been known so far [36]. The basic structural unit of flavonoids is diphenylpropane C_6_-C_3_-C_6_, the structure of which consists of two aromatic rings connected by three carbon atoms, which usually form an oxygenated heterocycle, and individual groups of flavonoids differ in the type of substituent in benzoic and phenyl rings, the position of the phenyl ring or the degree of oxidation of piran [35]. 

Phenolic acids (or phenolcarboxylic acids) are compounds made up of an aromatic ring, carboxylic groups and hydroxyl groups. Depending on the number of carbon atoms in the molecule, phenolic acids can be classified into several basic groups [37]. One encompasses the hydroxyl derivatives of benzoic acid, which consist of the carbon skeleton C_6_-C_1_ with methoxylations and hydroxylations at the aromatic ring. Derivatives of hydroxybenzoic acid include, for example, syringic acid and p-hydroxybenzoic acid. Hydroxycinnamic acids have a carbon skeleton consisting of nine carbon atoms (C_6_-C_3_) and a double hydrogen bond in the side chain. Naturally occurring phenol acids are mainly derivatives of cinnamic acid; examples are caffeic acid, chlorogenic acid, ferulic acid, ortho-meta- para-coumaric acid [38]. 

Lignans are dimeric structures formed by binding β-β′ between two units of phenylpropane. Individual groups of lignans differ in the degree of oxidation in the side chain and the substitution in aromatic formations [39]. Tannins are polyphenol derivatives of compounds that have a high molecular weight, i.e., from 500 Da to over 3000 Da. Tannins, in terms of chemical structure, are divided into hydrolyzing (galotanins) and non-hydrolyzing (condensed). Tannins with hydrolysis capacity are composed of gallic acid esters and ellagic acid glycosides. Such molecules are formed from shikimate, where the hydroxyl-sugar groups are esterified with phenolic acids [40]. Another group of compounds belonging to the polyphenolic compound group is stilbenes. They are compounds in which two phenyl molecules are connected by a S-S bonds [41]. They are present in a fairly small amount in the human diet, and their main representative is resveratrol (3,4′,5-trihydroxystilbene) in the form of both cis and trans [42]. The basic chemical structure of the selected polyphenols is shown in Figure 3.

### 3.2. Health Benefit Properties of Plant Polyphenolic Compounds

Numerous studies report that phenolic compounds have a pro-health effect on the human body [43]. Nevertheless, the real molecular interactions of polyphenols with biological systems remain mostly speculative. Potential confirmed mechanisms of action exist, as well as those that can occur in vivo. According to Fraga et al. [44], non-specific mechanisms include polyphenols as antioxidants (free radical scavenging and metal sequestration), as well as interactions with membranes. The specific mechanisms of action of polyphenols are interaction with enzymes, transcription factors and receptors. Polyphenols also work in biological processes through other mechanisms that are impossible to classify according to the above criteria [44].

Many scientific studies indicate that the use of a diet rich in polyphenols reduces the risk of chronic diseases [45]. The phenolic groups in polyphenolic compounds are able to accept electrons, and consequently, relatively stable phenoxyl radicals are formed. This causes disruptions in chain oxidation reactions in cellular components [41]. Due to their ability to inactivate or prevent the formation of reactive free radicals, polyphenolic compounds have antioxidant properties [46,47,48]. The consumption of compounds belonging to the discussed group has the ability to increase plasma antioxidant capacity. This may be due to the presence of reducing polyphenols and their metabolites in plasma, which may alter the concentrations of other reducing agents (sparing effects of polyphenols on other endogenous antioxidants). Another possibility is that they affect the degree of absorption of pro-oxidative substances contained in consumed food products – for example, iron [41]. Degenerative diseases such as cardiovascular disease, osteoporosis and cancer are associated with the aging process. Cancer is becoming an increasingly common disease. In 2020, 19.3 million new cancer cases were diagnosed globally. In addition, it has been noted 10.0 million cancer deaths. Lung cancer cases have increased by 2.21 million, breast cancer by 2.26 million, stomach by 1.089 million, liver by 0.96 million, and colon cancer by 1.93 million. In 2040, an estimated 28.4 million cancer cases are expected to happen globally [49]. Oxidative damage to cellular components, DNA, lipids and proteins accumulates with age and contributes to the degeneration of somatic cells and the pathogenesis of these diseases. By directly affecting reactive oxygen species or stimulating endogenous defense systems, antioxidant compounds from the diet can contribute to reducing these damages [45].

An important feature of polyphenols is their anti-inflammatory properties [29]. The mechanism of anti-inflammatory action of quercetin and rutin (flavonoids) mainly consists in inhibiting the activity of 5-lipoxygenase (5-LOX) and cyclooxygenase (COX, especially COX-2). These enzymes are involved in the synthesis of arachidonic acid prostaglandins and leukotrienes, which are mediators of the inflammatory response. Inhibition of these enzymes by flavonoids reduces the synthesis of, among others, PGE2 prostaglandin, leukotriene B4 and thromboxane A2, which contributes to inhibiting the influx of leukocytes, regulating the state of capillary tension and reducing inflammation. During the inflammatory reaction, large amounts of reactive oxygen form are formed, damaging collagen and blood vessel walls. Therefore, the combined antioxidant and anti-inflammatory effects of flavonoids can also improve the condition of blood vessels. The mechanism of the anti-inflammatory action of plants may vary depending on the chemical composition of the plant’s raw material. For example, the anti-inflammatory properties of curcumin result from impaired activation of NF-kB and decreased expression of many proteins involved in the inflammatory process (e.g., COX-2 and interleukins 8). In contrast, the anti-inflammatory activity of nettle compounds results from the presence of a leukocyte elastase inhibitor, cyclooxygenase and lipoxygenase, as well as a decrease in the level of inflammatory cytokines (TNF-α, II-β) and an effect on the level of NF-kB [50].

Polyphenol compounds exhibit a number of beneficial properties for health, repeatedly constituting a specific therapeutic potential [29,45,50]. Heretofore, among others, the anticancer, antidiabetic, neuro-protective, and cardio-protective effects of polyphenolic [36,41], and a positive impact on the course of osteoporosis, allergies and obesityhaves been proven [34]. The functional effects of polyphenolic compounds in relation to the aforementioned diseases are shown in Figure 4 [34]. A more detailed description of the effects of polyphenols on the cardiovascular system is described in Section 7 of this article.

The potential of dietary polyphenols to produce therapeutic effects can be at least partly attributed to the bi-directional link with the gut microbiome. This is due to the fact that polyphenolic compounds affect the composition of the intestinal microbiome in a way that leads to the preservation of homeostasis in the human body. In particular, the gut microbiome converts polyphenols into bioactive compounds that have therapeutic effects [36].

## 4. Polyphenolic Compounds as a Considerable Plant Component of the Daily Diet

Polyphenolic compounds are commonly found in plant foods (for example, vegetables, red fruits and grapes) and plant-based beverages (wine, green tea coffee) [51]. The content of these compounds varies, even when comparing varieties of the same species. For example, the formation of flavonol glycosides and flavonol glycosides depends on the access of light, and for this reason, their highest concentrations are in the leaves and outer parts of plants. In contrast, only small amounts are recorded in parts of plants that are underground [52]. Moreover, the sensory and nutritional properties of plant foods depend on the content of polyphenols [35]. Free phenolic acids are found mainly in vegetables and fruits [53]. In contrast, cereal products are mostly characterised by the presence of bound phenolic acids [54].

Table 1 provides information on the main dietary sources of the most important examples of polyphenolic compounds. 

It would be difficult to establish specific data on intake and recommended daily intake of polyphenolic compounds, also without omitting important insoluble polyphenols [35,64]. Currently, it is believed that polyphenolic compounds should be consumed together with five daily meals, and such intake is estimated at 0.5–1 g. According to the collected data, the most ingested polyphenols were flavonoids, and their consumption in Western countries varies between 50–800 mg. This value is higher in Eastern countries (the intake is even 2 g), and this is explained by the fact that the diet is much richer in vegetables and fruits [65].

A large number of epidemiological studies confirm the health-promoting effects of polyphenols or polyphenol-rich foods. However, more adequately powered, randomised, placebo-controlled human studies are still needed to fully understand the valuable potential of these compounds of plant origin [46].

## 5. Metabolism of Polyphenols by Gut Microbiota

Whether a bioactive substance will be effective in the human body depends mainly on its bioavailability [45,66,67]. It is not possible to completely use these ingested substances due to the fact that they are transformed or degraded during the digestive process [67]. In addition, interactions with other macromolecules (carbohydrates, proteins and lipids) affect the bioavailability of phenolic compounds [68,69]. The metabolism of polyphenolic compounds is related to their bioavailability—absorption, transport, distribution and retention in biological fluids, cells and tissues [70].

Research suggests that polyphenolic compounds are characterised by low bioavailability. This is due to several factors, such as interaction with the food matrix, the metabolic processes mediated by the liver (phase I and II metabolism), intestine and microbiota. Moreover, the metabolites of compounds produced in vivo may participate in their known biological activity—antioxidant or anti-inflammatory [66].

Gut microbiota (GM) is a heterogeneous and dynamic ecosystem of the gastrointestinal tract of the human body, which is populated by a huge number of microorganisms that interact with each other or the host. The gathering of all intestinal microorganism genes represents a genetic repertoire [71,72]. Due to the function of the key modulator of the human body [73,74], it has been proposed to be named an ‘essential organ’ of the human body [74]. GM has many important functions, such as the production of short-chain fatty acids, vitamins, and active metabolites, digestion of nutrients that are not able to be broken down by hosts, digestion and breakdown of plant components (including polyphenols), neutralization of potential mutagens and carcinogens, trophic impact on enterocytes and intestinal epithelium, interaction with the immune system, maintaining balance with pathogenic microorganisms and impact on the proper development and functioning of organs. Bacteria decompose bile salts into secondary salts acting as blockers on some pathogens, e.g., *Clostridium difficile*, whereby through the proper metabolism of bile acids, they support lipid adsorption and affect cholesterol homeostasis [50]. The main types of gastrointestinal microbiota are *Firmicutes*, *Bacteroides*, actinomycetes and Proteus [72]. 

The composition of GM in the entire digestive tract is not identical. In the initial proximal section of the small intestine, the types of bacteria present are similar to those in the stomach [75]. The diversity and number of bacteria increase in the distal part, from the duodenum to the ileum. This occurs with a gradual increase in pH. In this part of the gastrointestinal tract, the most common species are *Lactobacillus* and *Clostridium* of the genus Firmicutes, *Escherichia coli* of the genus Proteobacteria, as well as *Bacteroidetes* and Gram-negative elective anaerobes [76,77]. The gut microbiota is constantly changing throughout a person’s life. Its composition begins to form during fetal life and childbirth and then evolves depending on the way of feeding the child, and the diet followed in the subsequent years of life [78]. The body’s resistance and susceptibility to colonization by pathogenic microorganisms depend on the composition of the intestinal microbiota, with the ratio of *Bifidobacterium* to *Enterobacteriaceas*, which should be >1, being an indicator of whether it is normal for an adult [50].

The intestinal microbiome plays an important role in the metabolism of polyphenolic compounds supplied with food [79,80], especially those not digested in the upper gastrointestinal tract, which can be biotransformed by bacterial enzymes and subsequently absorbed. Due to the fact that there is a significant species diversity in the microbiome of different people, the profile of the emerging polyphenolic metabolites and their impact on the functioning of the body may differ. Polyphenols consumed with meals are metabolised by various pathways leading to the formation of many different phenolic derivatives. These low molecular weight derivatives differ in their current biological activity. For example, aglycones and oligomers are released by bacterial glycosidases and esterases and consequently increase their absorption. However, the released aglycones can inhibit the metabolism of other compounds by inhibiting the effect on GM activity. It often happens that only the metabolite of the polyphenolic compound created by the action of GM has a specific effect on the human body [81]. It should be noted once again that the bioavailability and bioeffectiveness of polyphenolic compounds are influenced by the amount of polyphenolic compounds supplied in the diet and the inter-individual differences in the composition of human microflora [75,81].

The structural complexity and polymerization of polyphenols affect their low absorption in the small intestine (less than 10%) [46]. Active compounds that have not been absorbed undergo several enzymatic changes as a result of interaction with the microflora of the colon lumen (such as hydrolysis, dehydroxylation, reduction, decarboxylation, demethylation and ring fission) [82]. As a result of these transformations, products with smaller molecules are formed [83]. The microbal metabolism of phenolic compounds in the GT is schematically illustrated in Figure 5 [83]. Certain classes of polyphenols, such as flavonols, isoflavones, flavones and athocyanins, are generally glycosylated. The linked sugar is often glucose or rhamnose. The number of sugars substituted may vary, as may the position of the polyphenol substitution [84]. The metabolism carried out by GM consists of a stage of degradation, during which aglycones and oligomers are released. This is done through glycosidases and esterases, which support their absorption [83]. The microflora then hydrolyzes the resulting aglycones into various aromatic acids. These structures are modified by the opening of the heterocycle at different points and at regions where flavonols produce hydroxy phenylacetic acids. Enzymes of microorganisms, glucuronidase and sulfatase, deconstruct the metabolism of phase II. They are then taken from the bile, and this allows them to be recaptured [83].

Flavones, flavones, flavanols [85], proanthocyanidins and phenolic acids have common metabolites such as hydroxyphenylpropionic acid, which are formed in the large intestine, while flavanols and dimers of ferulic acid—hydroxylated phenylacetic acid [70] are converted into gallic acid and ellagic acid by the GM. Gallic acid undergoes a decarboxylation and dihydroxylation reaction, while ellagic acid encounters dehydroxylation. After dihydroxylation of ellagic acid, metabolites of nasutin are formed. These compounds have two hydroxyl units removed. After the conversion of ellagic acid by lactone ring cleavage, decarboxylation and dehydroxylation reactions, it forms metabolites called ‘urolithins’ [86]. Through metabolic processes, anthocyanins convert into derivatives of benzoic acid, acetic aldehydes, hydroxylated benzaldehydes [70]. Herein, microbial decarboxylase enzymes change gallic acid, protocatechic acid and vanillic acid into pyrogalol, catechol and O-methylcatechol. This occurs when a free hydroxyl group is present at position 4 [86].

The intestinal microflora can convert resveratrol O-glucosides, such as transpiceid, into resveratrol aglycone by deglycosylation. After absorption, piceid and resveratrol are conjugated in the form of sulphate and glucuronide derivatives of the major circulating metabolites. The bacterial metabolites of trans-resveratrol are: dihydro-resveratrol, 3,4′-dihydroxy-trans-stilbene and 3,4′dihydroxydihydro-stilbene [86,87]. Additionally, hydroxybenzoic acid is the GM metabolite of pelargonidin glucoside, vanillic acid of peonidin glucoside, syringic acid of malvidin glucoside, methyl gallic acid of petunidin glucoside, and gallic acid of delphinidin glucoside [88]. Soy isoflavones can be converted to dihydrodaidzein, dihydrogenistein, 6′ -OH-O-desmethylagolensin and cis-4-OH-equol via anaerobic bacteria that reside in the distal small intestine and colon [89]. More complex metabolites of microorganisms (ellagitannins, lignans) are reabsorbed from the colon, then metabolised in the liver. Conjugated derivatives are excreted in the urine. Thus, the excretion from plasma and urine reflects the metabolism of polyphenols both in the liver and in the colon [90]. 

*E. coli, Bifidobacterium* sp., *Eubacterium* sp., *Bacteroides* sp. and *Lactobacillus* sp. hold particularly important roles in the metabolism of polyphenolic compounds [81]. The major bacteria that can metabolise anthocyanins are *Lactobacillus* spp. and *Bifidobacterium* spp. [91,92]. According to Selma et al., *Eubacterium* and *Clostridium* affect the metabolism of daidzein, quercetin, kaempferol, naringenin, catechin and epicatechin [79]. In addition, it was noted that *Enterobacter teriaceae* and *Lachnospiraceae* actively interact with rutin, and the type of final metabolite depends on the composition of GM [82]. Intestinal bacteria that may contribute to the metabolism of curcumin are *Escherichia coli* (the final product is tetrahydrocurcumin), *Escherichia fergusonii, Bifidobacteria longum*, *Bifidobacteria pseudocatenulaum, Enterococcus faecalis*, *Lactobacillus acidophilus* and *Lactobacillus casei* [93].

## 6. Effect of Polyphenols on the Composition of Gut Microbiota

More than 10,000 polyphenolic compounds [94] have been identified in plant products that increase the abundance of beneficial microorganisms. Because phenolic compounds can exert a prebiotic effect, it is crucial to understand their inhibitory or stimulating effect on beneficial or pathogenic bacteria. 

Polyphenolic compounds affect GM by influencing the growth and metabolism of bacteria and by interfering with the cellular function of the cell membrane. Moreover, a significant proportion of polyphenols act on GM by hindering biofilm formation and, notably, bacterial quorum sensing. For example, some classes of flavonoid polyphenols may affect *Staphylococcus* bacteria by decreasing bacterial helicase activity and increasing cytoplasm membrane permeability. In addition, a citrus extract containing flavanones and other isolated compounds from this group of secondary metabolites was found to contribute to hindering the formation of biofilm by inhibiting the quorum signal in which the lactone acyl-homoserine is mediated (these compounds inhibit its synthesis) [86]. Cranberry proanthocyanidins limited the motility and reduced the biofilm formation of *Pseudomonas aeruginosa* [95]. These compounds can also reduce the number of other pathogenic bacteria, such as *Helicobacter pylori, Escherichia coli* and *Clostridium perfringens* [96,97]. However, due to the structural diversity of polyphenol classes, the mechanisms of their antimicrobial activities have not yet been fully resolved. 

Gut microbiota integrity is key to the maintenance of gastrointestinal and overall body homeostasis. Growth as a biofilm promotes homeostasis at various mucosal surfaces, and disruptions of this mode of growth in such settings are detrimental to health. Microbial biofilms naturally colonise various surfaces of the body, including the gastrointestinal tract, the lungs, the vagina and the skin. Homeostatic maintenance of these complex microbial ecosystems is critical to health, and how their disruptions are a direct cause of pathophysiology. In the gut, the microbiome is made up of mixed communities of viruses, bacteria, fungi, and Eukarya that co-habit with mucus layers. Microbial colonization, diversity, and density vary along the length of the gastrointestinal tract, with the lowest numbers of microbes only forming scattered biofilm fragments in the stomach and upper gut, whereas a rich and dense microbial biofilm lines the large intestinal mucosa. Polyphenols contribute to maintaining the integrity of the gut microbiota biofilm. In general, polyphenolic compounds contribute to the growth and settlement of families of probiotic bacteria beneficial for human health (*Lactobacillaceae, Bifidobacteriaceae*) [98].

It is known that quercetin has a positive effect on the balance of bacteria in the *Firmicutes/Bacteroidetes* clusters ratio, restricting the growth of bacteria associated with obesity, such as *Erysipelotrichaceae, Bacillus* spp. and *Eubacterium cylindroides*. In addition, anthocyanins have been found to significantly stimulate the growth of *Bifidobacterium* spp., *Lactobacillus* and *Enterococcus* spp., suggesting that anthocyanins may positively select beneficial members of the intestinal microbial community [99]. For example, the polyphenolic compounds contained in tea (especially hydrolyzing tannins) are characterised by the ability to inhibit the development of numerous pathogens, including *Staphylococcus aureus* and *Esherichia coli* [78]. Green tea polyphenols also have a stimulating effect on the population growth of the *Firmicutes* and *Bacteroidetes* communities, as found in vitro and in animal studies. This brings about pro-health effects—reduction in *Firmicutes/Bacteroidetes* bacteria population and improvement in the ratio of *Prevotella/Bacteroides* [46]. Animal studies have shown that the administration of tea polyphenols restored the richness and diversity of the gut microbial population after previous administration of the antibiotic (cefixime), regulated gut microbial dysbiosis, and also increased the number of beneficial microbes such as *Lactobacillus, Akkermansia, Blautia, Roseburia* and *Eubacterium* [82]. According to research by Jaquet et al. [100], the consumption of a coffee preparation resulting from aqueous co-extraction of green and roasted coffee beans promotes metabolic activity and enhances the number of *Bifidobacterium* spp. population. Overall, coffee and tea consumption are considered to have a beneficial effect on the human body [100]. 

A crossover, controlled intervention study on the effects of moderate consumption of red wine has shown that this engenders improvement of blood pressure and lipid blood profiles and the composition of the gut microbiota. Compared to baseline, daily intake of red wine polyphenols for 4 weeks was found to significantly increase the number of groups of *Enterococcus, Prevotella, Bacteroides, Bifidobacterium, Bacteroides uniformis, Eggerthella lenta* and *Blautia coccoides-Eubacterium* rectale. The obtained results suggest possible probiotic benefits of wine [101]. Beyond the aforementioned, according to a randomised controlled trial by Vetrani et al. [102], a polyphenol-rich diet had an impact on the composition of the intestinal microflora wherein these modifications were associated with changes in glucose/lipid metabolism [102]. In related work, a proportional increase in *Akkermansia* spp. caused by dietary supplementation with cranberry extract, was found to be associated with an improvement in the characteristics of metabolic syndrome (induced by a diet high in fat and sucrose in an animal study) [103]. According to a clinical study, ingestion of bioactive compounds such as hesperidin and naringin (found in citrus fruits and orange juice) can improve intestinal microflora homeostasis (while enhancing blood biochemical parameters) by increasing the population of fecal *Bifidobacterium* spp. and *Lactobacillus* spp. [104]. The in vitro anti-Helicobacter pylori activity of citrus polyphenols such as hesperetin, naringenin, poncirin and diosmetin has also been demonstrated [105].

There are data from human studies that show changes in the diversity and composition of dominant bacterial communities in response to dietary supplementation with hormonal compounds in combination with functional foods. Isoflavones alone were noted to stimulate dominant microorganisms of the *Clostridium coccoides–Eubacterium rectale* cluster, *Lactobacillus-Enterococcus* group, *Faecalibacterium prausnitzii* subgroup and *Bifidobacterium genus* [106].

## 7. Polyphenols in the Cardiovascular System

The role of oxidative stress as a promoter of endothelial dysfunction [107] that, in turn, is a driver of early atherosclerosis and provides support to the anti-inflammatory and antioxidant strategy in terms of its prevention [108]. The antioxidant properties of polyphenolic compounds result from the presence of hydroxyl groups that can be readily oxidised [44]. This conversion causes scavenger activity towards reactive oxygen species. It takes place through the entrapment of free radicals into stable chemical complexes, thus preventing further reactions. This theory is currently the most validated one to explain the beneficial effects of polyphenols on various types of lifestyle diseases, including atherosclerosis. In addition to the one described inhibition of oxidative stress, polyphenolics also display indirect antioxidant activity through the activation of the nuclear transcription factor, erythroid 2-related factor 2 (Nrf2). This mechanism promotes endogenous antioxidant system development. In addition, it is probably also responsible for the polyphenol-mediated redox homeostasis of cells [109]. The anti-inflammatory properties of phenolic compounds are strongly connected with the modulation of oxidative stress and the maintenance of cellular redox homeostasis [110]. 

Many mechanisms, most of which are mediated by the inhibition of the nuclear factor kappa B (NF-kB), are responsible for the anti-inflammatory properties of phenolics. This group of compounds can decrease the cellular production of pro-inflammatory mediators [111]. Moreover, polyphenols are able to inhibit the expression of adhesion molecules [112], thus impairing the chemotaxis of monocytes within the inflamed tissues. Polyphenolic compounds have been reported to slow down smooth muscle cell proliferation by interfering with mitogen-activated protein kinase (MAPK) activity. In addition, protocatechuic acid has been shown to mitigate monocyte adhesion and blunt atherosclerosis in ApoE-/- mice [113].

An important mechanism responsible for the positive effect of polyphenols on the cardiovascular system is connected with lipid metabolism, the impairment of which represents a causative factor of atherosclerosis [107]. The decrease of total and low-density lipoprotein cholesterol (LDL-C) following the intake of polyphenolic compounds is possibly related to processes occurring at the hepatic and intestinal levels. The reduction of cholesterol synthesis, the increase of LDL receptor expression and activity, as well as the increase of the cholesterol transporters ATP-Binding Cassette G5/ATP-Binding Cassette G8 expression have been described in vivo models [114]. In addition, the ability of polyphenols to displace cholesterol from intestinal micelles results in greater elimination of cholesterol in the feces in vivo [115]. The effect on triglyceride plasma level is possibly related to the reduction of apolipoprotein B48 and apolipoprotein B100 production in the liver and intestine [116]. In contrast, the increase of high-density lipoprotein cholesterol HDL-C level is related to an increase in apolipoprotein A–I synthesis, as indicated in cultured hepatic or intestinal cells exposed to polyphenols action [117].

Phenolic compounds can also positively influence endothelial function, the impairment of which is one of the most significant factors causing atherosclerosis. High intake of these compounds (significantly higher than in typical diets) has been shown to enhance flow-mediated dilation (FMD) in patients. The mechanism responsible for FMD improvement likely relies on the increase of NO synthase activity, as indicated in vitro studies [118] and human studies [119]. Polyphenols can exert a beneficial effect on the vascular and endothelial functions through several additional mechanisms, e.g., a normalization of the local angiotensin system and an inhibition of oxidative stress, by preventing the expression of pro-oxidant enzymes such as NADPH oxidase and cyclooxygenases (COXs) [120].

Furthermore, the cardiovascular benefit of polyphenol intake may be ascribed to peculiar pharmacokinetic characteristics. These compounds reach the distal section of the gastrointestinal system unchanged, and after modification by the gut microbiota, they then exert a prebiotic-like activity. They are able to cause the selective growth of beneficial bacteria with the inhibition of harmful strains, and this effect may be responsible for the amelioration of markers of CVD [121].

## 8. The Impact of Polyphenols and Their Metabolites on the Mechanisms and Factors Causing Atherosclerosis

Atherosclerosis is a chronic inflammatory disorder of medium and large arteries and an underlying cause of cardiovascular disease. As previously stated, one of the main risk factors causing atherosclerosis is the type and quality of food that the population eats. According to the dietary guidelines presented by the US Health and Human Services, only about 30% of the American population meets the dietary recommendations for fruit consumption and less than 20% for vegetables [122]. Moreover, almost 90% of Americans overuse salt, and 70% consume excessive amounts of saturated fats and sugars. Consumption of fruits and vegetables is very important due to the presence of polyphenols with a wide range of health-promoting properties, especially in inflammatory diseases like atherosclerosis [113].

The poor dietary choices made by many individuals have profound pathological effects and therefore contribute to the development of cardiovascular diseases. Current treatments for CVD, such as optimised statin therapy, are associated with considerable residual risk and several side effects in some patients.

The outcomes of research on the identification of alternative pharmaceutical agents for the treatment of atherosclerosis are mostly disappointing, with many promising leads failing at the clinical level. Reach in polyphenols food products with health benefits beyond their nutritional value represent promising agents in the prevention of CVD or as an additional therapy with current treatments. Polyphenols play a role in reducing ROS, inflammatory processes such as monocyte adhesion and vascular smooth muscle cell proliferation and migration, all of which are key events in atherosclerosis. This section will highlight the potential of several plant polyphenols and polyphenol reach products, including fruits, vegetables, teas and oils, as anti-CVD therapies based on clinical and pre-clinical mechanism-based studies.

### 8.1. Cacao and Green Tea Flavonols

Catechins are a class of flavanols mainly present in cocoa and green tea. This group of phytochemicals attracts interest because of their antioxidant properties and their ability to inhibit the secretion of pro-inflammatory compounds from activated endothelial cells [123]. In evaluating their effectiveness in reducing cardiovascular diseases, e.g., atherosclerosis, a trial involving 27 healthy people receiving a cacao flavanol-rich diet for 5 days revealed an increase in NO production and improved vasodilation [119]. In addition, as end-stage renal disease (ESRD) sufferers have an increased risk of CVD due to impairment of their vascular function, a trial involving 57 ESRD participants found improved vascular function (increased flow-mediated dilation (FMD), and reduced blood pressure [124]) after 30 days of cocoa flavanol-rich dietary supplementation. 

A high risk of CVD occurs in individuals with obesity. However, an intake of 814 mg/day of catechins for 4 weeks increased vasodilation in 30 overweight patients. This came without changes in HDL, LDL-C and total cholesterol (TC) levels between those patients and the control group [125]. Other studies have shown that a cocoa-enriched calorie-restricted diet in 50 healthy people reduced their oxidised LDL (oxLDL) levels after one month when compared to calorie restriction only [126]. In addition, in human intervention trials in normo- and hypercholesterolemic subjects, the intake of 13 g/day cocoa powder for 4 weeks had beneficial effects on LDL and HDL cholesterol and oxidised LDL concentrations in plasma, especially in subjects with high LDL level [127]. 

In individuals at high risk of CVD, the intake of cocoa flavonols has shown to modulate levels of inflammatory mediators and serum levels of the leukocyte-endothelial cell adhesion molecules P-selectin and soluble intercellular adhesion molecule (sICAM-1). Other human, 7-month consuming studies conducted on hemodialysis patients have shown that the catechins administration (455 mg per day in the first group), as well as four cups of green tea (second group), decreased atherosclerotic pro-inflammatory factors such as monocyte chemoattractant protein 1 (MCP-1), TNF-α, sICAM-1 and CRP with respect to a control group. These anti-inflammatory effects may contribute to the overall benefits of cocoa and decaffeinated green tea extracts consumption against atherosclerosis [128]. Catechins also have antioxidant and anti-inflammatory effects on the endothelium. A reduction in the concentration of CRP protein and monocyte chemotactic protein was shown in people who consumed 5–6 cups of green tea per day for 14 days. Taking 580 mg of green tea catechins a day contributed to a decrease in the blood concentration of pro-inflammatory cytokines—TNF-α and IL-6—as well as the concentration of 8-hydroxy-2-deoxyguanosine, one of the markers of damage to the DNA structure by reactive oxygen species produced in the inflammatory reaction [129].

In the past decade, three large-scale studies have been conducted to investigate the potential cardioprotective effects of flavanols—PREDIMED [130], Cocoa, Cognition and Aging study (CoCoA) [131] and Flaviola Health study [132]. A test population of 7172 people participated in the largest of these, PREDIMED—a 4-year observational study. The patients were administered various doses of flavanol, from 90 mg to 263 mg per day. The findings confirmed a significant association between increased flavanol intake and reduced CVD risk, even after consideration of other risk factors (e.g., lipid-lowering therapies, other nutrients etc.) [130]. In the second study—CoCoA (Cocoa, Cognition and Aging), 90 elderly patients consumed either a low (48 mg per day), intermediate (520 mg per day) or high (993 mg per day) flavanol dose for 8 weeks [131]. The results of the study showed that individuals who had received either the high or intermediate daily dose of flavanols had significantly reduced insulin resistance, better blood pressure parameters, as well as a reduction in the amount of lipid peroxidation [131]. In the (Flaviola Health study, similar observations were found. During this study, 100 healthy women and men without a history of CVD received, for 1 month, 900 mg of cocoa flavanol per day [132]. Accordingly, flavanol supplementation significantly increased FMD and HDL levels, and simultaneously reduced blood pressure, arterial stiffness, as well as total and LDL-C levels [132].

Animal studies also confirm the beneficial effects of cacao flavonols. Kurosawa et al. [133] administered these compounds in the form of a cacao liquor polyphenols (CLP), in the spontaneous familial hypercholesterolemic model, to Kurosawa and Kusanagi-hypercholesterolemic (KHC) rabbits. After 1 month of dietary administration of CLP at 1% to the KHC rabbits, the authors observed that the plasma concentration of thiobarbituric acid reactive substances (a lipid-peroxidation index) was significantly decreased. Moreover, the antioxidative effect of CLP on LDL was observed from 2 to 4 months after the start of administration. In addition, the area of atherosclerotic lesions in the aorta in the CLP group was significantly smaller than that in the control group, and the tissue cholesterol and thiobarbituric acid reactive substances concentrations were lower in the CLP group than in the control group. Hence, the anti-atherosclerotic effect of CLP was demonstrated both rheologically and histopathologically. Finally, the authors confirmed that the antioxidative effect of CLP was superior to those of the well-known antioxidative substances (probucol, vitamin E and vitamin C). Therefore, CLP suppressed the generation of atherosclerosis, and its antioxidative effect appeared to have an important role in its anti-atherosclerotic activity. Overall, it can be said that cacao flavonols have an activity that causes the maintenance of healthy blood vessels, e.g., inhibition of vasodilative activity through controlling the levels of NO and eicosanoids [132], regulation of cytokine production, and an antioxidative effect on LDL. These effects might contribute to the anti-atherosclerotic effect [133].

### 8.2. Resveratrol

Resveratrol is a natural polyphenol derived from wine that has powerful antioxidant, anti-inflammatory and anti-aging effects. Numerous studies have shown that this compound has protective effects against cardiovascular diseases. Timmers et al. [134] found that resveratrol in obese people had a metabolic effect similar to that of caloric restriction after 30 days of supplementation. In particular, blood pressure, sleep and resting metabolic rates, triglycerides, glucose and markers of inflammation were all lowered. There was a reduction in systolic and diastolic blood pressure and an improvement in lipid metabolism indices in people with type 2 diabetes after three months of daily administration of 250 mg of resveratrol. In further studies [135], a similar set was used, with the addition of resveratrol in the diet of obese patients. This time, however, the resveratrol treatment was longer and combined with epigallocatechin-3-gallate (EGCG). The intention was to seek synergies between polyphenolic compounds. However, the effects were modest: supplementation improved only several biochemical indicators, e.g., the blood lipid profile and skeletal muscle capacity. However, changing these biochemical markers did not translate into clinically significant improvements in insulin sensitivity, which is very much expected in patients with obesity. Another study of dietary resveratrol in humans was conducted within one year [136]. The findings confirmed that resveratrol downregulated the expression of several pro-inflammatory cytokines. The levels of TNF-α, C-C motif chemokine ligand 3 (CCL3), interleukin- 1-beta (IL-1-β), interleukin-8 (IL-8), and chemokine (C-X-C motif) ligand 2 (CXCL2) were diminished. The treatment also modulated the expression of inflammation-related micro-RNAs. However, the effects were not very impressive—typically, less than two-fold changes were detected [137].

Recent studies have shown that resveratrol has the ability to inhibit mTORC1 activation [i]. mTOR (the mammalian target of rapamycin) belongs to the PIKK family (phosphatidylinositol kinase-related kinase), and is a serine/threonine-specific protein kinase that plays a key role in many physiological processes. mTOR inhibition has been suggested to have a helpful effect on atherosclerosis, heart failure and myocardial hypertrophy. In this context, mTOR inhibition is profitable in the treatment of many cardiovascular diseases, including atherosclerosis [138].

Substances focused on mTORC1 inhibition reduce the atherosclerotic process mainly by correcting the function of the endothelial layer and by inducing autophagy. This process diminishes the content of macrophages in plaques and causes the efflux of cholesterol from plaques. mTORC1 inhibition also reduces the formation of foam cells, which are fat-filled macrophages that engulf cholesterol esters and LDL [139]. In addition to the effective treatment of CVD, mTOR inhibitors have proven to be effective therapies for hypertensive heart diseases, such as hypertension, heart failure and cardiac hypertrophy.

Other studies have confirmed that resveratrol inhibits mTOR [140] in the vascular system, including smooth muscle cells and endothelial cells. Moreover, it contributes an inhibitory effect on oxLDL-induced smooth muscle cell proliferation and age-related endothelial dysfunction, as well as to endothelial damage caused by oxidative stress. The oxLDL-induced proliferation of smooth muscle cells is believed to be one of the major factors contributing to atherosclerosis. It develops fibroatherosclerotic plaques, mainly through the activation of PI3K/Akt signaling. In addition, resveratrol was found to significantly reduce palmitic acid (PA)-induced reactive oxygen species (ROS) generation and improve endothelial dysfunction by inducing autophagy via the AMP-activated protein kinase AMPK-mTOR pathway in human aortic endothelial cells. The effects exerted by mTOR inhibition can contribute to its therapeutic effects in ameliorating atherosclerosis in animal models of atherosclerosis [141]. Due to its outstanding ability to activate SIRT1 (NAD-dependent protein deacetylase sirtuin-1) and AMPK, the protective effects of resveratrol on the cardiovascular system may also involve both targets.

### 8.3. Curcumin

Studies have shown that curcumin prevents atherosclerosis in several animal models, doing so through multiple mechanisms, e.g., antioxidant, anti-aging and anti-inflammatory effects [138]. For example, a human study supplementing with 1g of curcuminoids daily for 8 weeks increased patient superoxide dismutase levels while lowering plasma malondialdehyde concentrations, indicating reduced oxidative stress. In addition, those receiving curcuminoids had decreased plasma CRP levels [142]. Moreover, a meta-analysis of clinical trials also found reduced CRP levels with curcumin, although no effect was seen on plasma TC, HDL, and LDL-C [143]. Curcumin also inhibits mTOR activation in the vascular system. This compound has been reported to protect endothelial cells from oxidative stress-induced damage and apoptosis by promoting autophagy by inhibiting mTOR [138]. Recent results confirm that the nicotinate-curcumin hybrid impedes the formation of macrophage-derived foam cells by enhancing autophagy. This effect may be due to its mTOR inhibitory effects [144]. Similarly, hydroxyacetylated curcumin has a beneficial effect in delaying the formation of foam cells.

### 8.4. Quercetin

Quercetin is a flavonoid common in fruits and vegetables. In randomised clinical trials, quercetin supplementation with 150 mg daily for 6 weeks reduced blood oxidised LDL in adults at high risk of cardiovascular disease. Interesting results were obtained in adult smokers who supplemented with 100 mg of quercetin daily for 10 weeks. In these people, quercetin significantly improved the plasma lipid profile by lowering total and LDL cholesterol levels and increasing the HDL fraction levels in the blood [145].

Quercetin’s mTOR inhibitory activity may partly contribute to its anti-atherosclerotic effect, which has been observed in animal models and cultured cells of atherosclerosis [138]. A recent study has shown that this flavonoid delays angiogenesis via inhibiting the vascular endothelial growth factor receptor 2 (VEGFR-2)-dependent protein kinase AKT/mTOR pathway [146]. In addition, research conducted by Liu et al. [147] confirmed that quercetin attenuates the rise of lipid levels in the liver by inhibiting mTOR (large lipid amounts in the liver and macrophages initiate atherosclerosis).

Quercetin also activates the AMPK pathway in vascular smooth muscle cells and contributes to the inhibition of induced contraction of the rat aorta. Taking into account this mechanism, it is likely that quercetin restricts mTOR generation via the AMPK-dependent pathway. This flavonoid also increases the activation of the SIRT1 in oxLDL-stimulated endothelial cells, which may also induce an mTOR-inhibiting effect [148]. 

A study by Shen et al. [149] showed that in ApoE-/- mice treated with quercetin at a dose of 1.5 mg per day, TC and TG levels were significantly decreased after 14 weeks. The atherosclerotic-protective action of quercetin put forward by Loke et al. [150] resulted from observations of improvements in NO availability by increased nitric oxide synthase (eNOS) activity and heme-oxygenase-1 (HO-1), as well as by a reduction in leukotriene B4 and LDL oxidation [150]. Leukotriene B4 is an agonist of inflammatory responses connecting with TNF-α and interleukins and the recruitment of neutrophils and monocytes [113]. Shen et al. [149] also reported an increase in eNOS and HO-1 action as the main mechanism connected with atherosclerosis reduction via quercetin. Of note, HO-1 mediates the rate-limiting phase of heme degradation. Moreover, HO-1 prevents atherosclerosis by affecting bilirubin, which reduces lipid peroxidation. Heme-oxygenase-1 also inhibits the development of the disease in mice in a lipid-independent way [113].

Quercetin-3-glucuronide has been reported to decrease the formation of macrophage foam cells by restricting the expression of CD36 macrophages and scavenger receptor A1 (SR-A1) in RAW 264.7 cells. This flavonoid is known to accumulate in atherosclerotic lesions in the human aorta (in particular, in the macrophage-derived foam cells). In addition, metabolites of quercetin might exhibit anti-atherosclerotic effects in injured/inflamed arteries with activated macrophages [151].

### 8.5. Protocatechuic Acid

Protocatechuic acid (PCA) is a bioactive compound present in medicinal herbs, vegetables, fruits and spices. It is the main metabolite of anthocyanins produced by the intestinal microflora [152]. PCAs have been shown to have promising anti-atherosclerotic effects at physiologically achievable concentrations. The protocatechuic acid derivative can accelerate cholesterol efflux in MPM loaded with AcLDL or THP-1 macrophages. Furthermore, PCA can inhibit intercellular adhesion molecule 1-dependent (ICAM1-dependent) monocyte adhesion and vascular cell adhesion molecule 1 (VCAM-1), activated human umbilical vein endothelial cells (HUVEC), as well as CCL2-mediated monocyte transmigration, thereby reducing the development of atherosclerosis in ApoE-/- mice. PCA has also been demonstrated to have an inhibitory effect on VSMC proliferation (induced by oleic acid) by activating AMPK and arresting the cell cycle in the G0/G1 phase in the A7r5 smooth muscle cell line [152]. Therefore, it can also be explored as a potential new molecule for the prevention and treatment of atherosclerosis.

### 8.6. Trimethoxycinnamic Acid

α-Asarone is a biologically active compound from the class of phenylpropanoids. It shows hypocholesterolemic action. 2,4,5-trimethoxycinnamic acid, the non-toxic metabolite of asarone, has been found to represent its main pharmacological properties, e.g., lowering total blood cholesterol, high- and low-density lipoprotein cholesterol in hypercholesterolaemic rats [151]. Therefore, α-asarone is a subject of further research as a hypocholesterolemic and cardiovascular protective agent.

### 8.7. Gallic Acid

Gallic acid (GA) is an anthocyanin metabolite that has been found to improve atherosclerosis through antihypertensive and vasopressor action. This phenolic acid can increase nitric oxide levels by increasing endothelial eNOS phosphorylation in RAW264.7 macrophages. In addition, its administration inhibited angiotensin-I converting enzyme (ACE), causing blood pressure reduction in spontaneously hypertensive rats that were comparable to captopril [153]. These results indicate that gallic acid exhibits multiple therapeutic properties and has the potential to prevent atherosclerosis.

### 8.8. Equol

Equol is an isoflavone. This compound is formed as a result of the processing of daidzein by the intestinal flora. It has been suggested that equol may slow down the development of atherosclerosis by attenuating endoplasmic reticulum stress and, in part, by activating the Nrf2 signaling pathway [151]. A clinical study in Japanese men showed that equol might have high atherosclerotic-protective properties [154]. Evidence from other studies and short-term randomised controlled trials show equol has an antiatherosclerotic effect, improves arterial stiffness and can prevent ischemic heart disease [155]. Therefore, the use of equol has developed promise in the field of cardiovascular disease prevention.

### 8.9. Mediterranean Diet 

The Mediterranean diet is the gold standard for polyphenol dietary intake; therefore, it has been subjected to much research [123]. The PREDIMED study was the largest trial investigating the potential cardiovascular protective effects of this diet [130]. Initial analysis of the study, conducted on over 7000 Spanish people, found that after approximately 5 years, those with the highest polyphenol intake had a 37% lower relative risk of all-cause mortality when compared to those with the lowest polyphenol intake [156]. Moreover, further research on 200 high-risk patients found that those on the Mediterranean diet had reduced blood pressure after 1 year compared to those on a control diet [157]. In this work, a positive association between total polyphenol intake and plasma NO level was noted, suggesting a possible mechanism by which polyphenols induced vasodilation and reduced the risk of CVD [158]. Improvement in blood pressure and reduction in plasma TG were also found in a subset of 573 elderly participants of the project after a 5-year follow-up [158]. The PREDIMED study indicated an association between the Mediterranean diet and the cardiovascular protective effects of polyphenols through vascular function improvement and blood pressure reduction. Of note, all of the participants of the PREDIMED study were recruited from a single country (Spain), and therefore, differences in lifestyle between countries may alter the outcome of the study. 

A study on the components of the Mediterranean diet conducted by Jungeström et al. [159] included 1139 high-risk participants and revealed reduced plasma levels of several inflammatory biomarkers related to atherosclerosis with polyphenol intake [159]. Another study on 78 obese patients who were randomised to receive one of the following diets for 8 weeks; low omega-3 polyunsaturated fatty acids (PUFAs), low polyphenol, high omega-3 PUFAs, low polyphenol, low omega-3 PUFAs, high polyphenol, or high omega-3 PUFAs high polyphenol, but found no changes in their blood pressure or total and LDL cholesterol levels. Those on the high polyphenol diets did have reduced TG levels, and their HDL plasma concentrations were also lowered [160]. Upon further analysis of this study, an inverse correlation between LDL-C levels and the intake of gallic acid was suggested. 

The Effect of Olive Oils on Oxidative Damage in European Populations (EUROLIVE) randomised clinical trial found a high positive correlation between the intake of polyphenols in the form of olive oil and serum HDL levels in 200 healthy men after three weeks of a rich-olive oil diet. Furthermore, oxidative stress markers decreased with increased polyphenol concentrations, while TG levels were reduced in all patients [161]. This decrease in oxidative stress markers was confirmed in a later study on 117 individuals with metabolic syndrome [123]. 

The consumption of extra virgin olive oil is associated with a reduction in inflammatory biomarkers and molecules implicated in atherosclerosis as well as CVD incidence and mortality as well as other complications such as heart failure and atrial fibrillation. These anti-inflammatory and cardioprotective effects of extra virgin olive oil are mostly attributable to its high content of antioxidants, e.g., monounsaturated fatty acids, tocopherols and polyphenols. The cardioprotective properties of the Mediterranean Diet were demonstrated for the first time in the Seven Country Study [113]. Rosenblat et al. found that extra virgin olive oil with green tea polyphenols had significant effects on plaque reduction throughout the entire aorta of about 20%, compared to extra virgin olive oil alone (11%) [162]. Eilertsen discovered that extra virgin olive oil with and without green tea polyphenols can reduce oxLDL and lipid peroxidation [163]. Overall, supplementing oils into the diet, especially olive oil, reduces atherosclerotic plaque; the primary mechanism of interest for this result is the improvement of antioxidant capacity. However, there is a limited amount of available randomised controlled trials.

### 8.10. Other Polyphenol-Rich Foods

Grapes and pomegranates have been studied for their potential beneficial effects in ApoE-/- mouse models of atherosclerosis and CVD. The most abundant polyphenols in grapes are anthocyanins, flavanals, flavanols and resveratrol [113]. Pomegranates are also rich in anthocyanins, flavanols and ellagitannins, but additionally contain punicalagin, which is specific for pomegranates [113]. The pomegranate juice, flowers and byproducts caused significant changes in plaque after their supplementation. Aviram et al. [164] noted a 44% reduction of plaque through juice consumption, while Kaplan et al. reported a 17% reduction [165]. Aviram et al. additionally confirmed a 39% reduction with byproduct powder, a 38% reduction with byproduct liquid, and an amazing 70% reduction of atherosclerotic plaque with pomegranate flower intake [164]. Pomegranate flower preparations were additionally able to reduce serum glucose levels [113]. Rosenblat et al. [166] reported that pomegranate byproduct reduced 57% of plaque. In studies involving red wine grape pomace, a significant reduction in lesion size was seen [113]. Herein, grape powder polyphenols reduced the lesion size by 41%. A reduction in plaque in the thoracic aorta was also observed with both white (30%) and red (62%) dealcoholised wine. Additionally, the red wine reduced plaque by 16% in the aortic root. Overall, it can be concluded that pomegranate flowers and grape extract can reduce TC and TG. 

In many studies conducted on ApoE-/- mouse models, the attributed cause of plaque reduction was due to decreases in macrophage oxidative stress, oxLDL uptake, and reduced lipid peroxide concentration [164]. Here, the peroxide concentration is low can be attributed to the high free radical scavenging ability of the grape and pomegranate polyphenols. In addition, polyphenols interfere with oxLDL binding to macrophage scavenger receptors and, in this way, reduce cholesterol uptake [164]. Rosenblat et al. [166] also found that pomegranate polyphenols significantly increased glutathione levels. This action contributes to a reduction in the potential of cells to oxidise LDL cholesterol. 

Peluzio et al. [167] discovered that grape extract (enriched with vitamin E) increases the expression of LDL receptors in the liver. This activity leads to increased cholesterol uptake by the liver, thereby reducing circulating cholesterol levels. The authors also noted increased fecal excretion of cholesterol and triacylglycerols, hence supporting their conviction that higher cholesterol removal is one of the main benefits of treatment using pomegranate phenolic compounds [167]. In related studies, dealcoholised wine was found to reduce atherosclerotic plaque by inhibiting VCAM-1 and ICAM-1 adhesion molecule pathways, including NF-κB, MAPK, interferon type 1 (IFN-1), and IL-1β. Thus, grapes and pomegranate show promising results in ameliorating atherosclerosis by reducing oxidative stress by decreasing oxLDL uptake by macrophages and lipid peroxidation [113].

Shema-Didi et al. [168] found a reduction in blood pressure in 101 hemodialysis patients with a high risk for CVD who were treated with pomegranate juice three times a week for 1 year. During this study, there a correlation was also noted between the length of juice intake and time and improvements in HDL and TG levels. Furthermore, the subset of participants who had pathologically high TG and HDL levels at the start of the research showed significant improvements in these factors [168]. 

Many other fruits, for example, apples, lychee and plums, also have positive effects on atherosclerosis prevention. Their polyphenols probably act through the reduction of oxidative stress by reduction in adhesion molecules, as well as through inhibition of inflammatory pathways and enhancement of antioxidant capacity. Apples contain quercetin, flavonols, procyanidins, and phenolic acids. Lychee is a source of many polyphenols, e.g., catechins, anthocyanins, and procyanidins. Plums have a slightly varied polyphenol profile, and contain neochlorogenic and chlorogenic acids, which improve antioxidant capacity and reduce LDL [113]. The apple studies conducted on ApoE-/- mice showed that these fruits could reduce atherosclerosis through a large variety of metabolic actions, including increased superoxide dismutase (SOD) and glutathione peroxidase (GPx) activity [169], peroxisome proliferator-activated receptor α (PPARα) and Nrf2, reduced plasma uric acid concentration and decreased circulating cholesterol in serum [170]. Additionally, Xu et al. [169] showed reduced VCAM-1 level by apple polyphenols, similar to dealcoholised wines. The lychee study proposed that increased NO production attenuated atherosclerosis, while the plum study suggested that its intake increased serum amyloid P-component (SAP) levels that reduce inflammation [113]. 

Research on vegetable polyphenols using ApoE-/- mice was conducted by Lin et al. These authors studied chicory [171], while Joo et al. [172] studied anthocyanins from red Chinese cabbage. A lipid profile analysis showed that red Chinese cabbage had beneficial effects, specifically by reducing the TC and LDL/VLDL levels [172], while the chicory study showed reduced cholesterol in the aorta [171]. Chicory was reported to decrease plaque by lowering TC and activating ATP-binding cassette transporter 1 (ABCA-1) and ATP-binding cassette sub-family G member 1, which are pivotal players in HDL-dependent cholesterol efflux [171]. Anthocyanins from red Chinese cabbage were also noted to reduce VCAM-1 and improve antioxidant capacity and lipid metabolism [172].

It seems that these effects result from the action of the primary polyphenols in cruciferous vegetables (kaempferol, isorhamnetin, quercetin and caffeic acid derivatives). These polyphenols can promote eNOS activity and HO-1 expression to reduce oxidative stress and plaque accumulation.

## 9. The Anti-Atherosclerosis Therapeutic Potential of Polyphenol by Gut Microbiota Modulation

Polyphenols are thought to prevent the development of chronic diet-related diseases; however, most of them can not be absorbed directly by the small intestine. Therefore, their bioavailability and impact on the host mostly depend on their conversion [w]. Gut microbiota (GM) plays a crucial role in converting dietary polyphenols into absorbable bioactive substances. Indeed, some intestinal metabolites derived from natural polyphenol products have more biological activities than the chemical compounds from which they originate.

Several studies have shown that the antiatherosclerotic effect of flavonoids is the result of the activity of metabolites formed under the influence of gut microbiota. For example, protocatechuic acid is a metabolite of anthocyanins that causes an anti-atherogenic effect through miRNA-10b-ABCA1/ABCG1-cholesterol efflux signaling cascade [173]. Food intake is a two-way street, however, and studies have indicated that the intake of anthocyanins remarkably remodeled the “pro-atherogenic” GM community to the desired state through increasing relative taxa *Lactobacillus, Akkermansia, Bifidobacterium,* and *Roseburia*. In contrast, the content of Prevotellaceae members has decreased. Alongside the GM modulation, protocatechuic acid regulated the NF-κB arterial inflammatory pathway and hepatic lipid metabolism. Consistently, aberrant morphological changes in the aorta, intestine, and liver, as well as deregulated endothelial function biomarkers (VCAM-1, TNFα), were also found to be mitigated due to flavonoid intake [174]. Thus, the administration of anthocyanins significantly reduced atherosclerosis in rats. Still, how these results can be translated into the ‘real life’ with the ever-changing gut microbiome has not been fully understood. Chen et al. [175] noted that polymethoxyflavones (PMFs) from citrus peels exhibit comprehensive biochemical functions, including antioxidant, anti-inflammatory, etc. Moreover, PMF intake modulated the GM composition to a healthier phenotype, basically increasing *Lactobacillus, Bacteroides, Akkermansia,* and *Bifidobacterium* presence. In this activity, butyrate-producing microbes and TMA-producing microbes were specifically increased and decreased, respectively, under PMFs action. PMFs also inhibited the biosynthetic pathway of trimethylamine N-oxide (TMAO) generation by lessening the transformation of L-carnitine to TMAO. 

In related work, nobiletin, a compound belonging to polymethoxyflavones was found to suppress NF-κB and mitogen-activated protein kinase/extracellular signal-regulated kinase signaling pathways to avoid TMAO-induced vascular inflammation [176]. Moreover, other PMFs blocked L-carnitine-induced vascular inflammation via decreasing VCAM-1, TNFα, and E-selectin. In addition, macrophage foam cell formation was suppressed by PMFs [175]. Another study involving humans looked at quercetin, and the use of this flavonoid was found to inhibit platelet aggregation and thrombus formation. Similarly, in animal models, its use drastically improved lipid metabolism and inflammation.

Quercetin has also been found to influence the composition of the gut microbiome. This led to the association of the genera Anaerovibrio and Phascolarctobacterium as specific microflora characteristics of its therapy [177]. In researching quercetin, 32 metabolic signatures were identified as participating in quercetin actions, among them, the primary bile acid biosynthesis pathway. Thus, quercetin might be acting directly or indirectly to encourage GM metabolite bile acid. As previously highlighted, bile acids act to maintain host metabolic activities such as adipose tissue browning, insulin sensitivity, intestinal barrier integrity, etc. [178]. In other work, tryptophan and sucrose metabolic pathways were identified during in vivo studies as quercitin’s area of action.

Ferulic acid is a polyphenol with antioxidant and anti-inflammatory activities. This phytonutrient can modulate GM composition by decreasing the ratio of *Firmicutes/Bacteroidetes*. In research involving ferulic acid, it was found to decrease tryptophan plasma concentration that was elevated in high-fat diet-fed ApoE-/- mice [179]. Other research has indicated that pomegranate juice intake can abolish acrolein-induced GM dysbiosis by rebalancing *Firmicutes/Bacteroidetes* ratio and reducing the concentration of the order *Clostridiales* and the genera *Coprococcus* and *Dehalobacteria*. This polyphenol-rich drink was discovered to inhibit acrolein-induced lipid accumulation, peroxidation, and oxidative stress by suppressing essential lipid biosynthetic pathways of 3-hydroxy-3-methylglutaryl-COA reductase, diacylglycerol acyltransferase 1, and Sterol-regulatory element-binding protein maturation in in vivo and in vitro atherosclerotic models [180]. Moreover, Chitin-glucan plus polyphenol-rich pomegranate peel extract was seen to modulate Alistipes and *Lactobacillu’s* relative abundance and also improve hepatic lipids, decrease inflammation, and ameliorate endothelial dysfunction through eNOS activation in high fat diet-fed ApoE-/- mice [181]. In addition, resveratrol was found to be able to promote the beneficial microbes, e.g., *Bacteroides, Bifidobacterium, Lactobacillus,* and *Akkermansia*. In research, this action elevated bile salt hydrolase activity through cytochrome P450 27A1 and hepatic bile acid neosynthesis. In this way, bile acid deconjugation and fecal excretion are promoted to contribute to cholesterol homeostasis [182]. 

In related work, tea polyphenols reduced plaque/lumen area and enhanced the relative abundance of *Bifidobacterium* population in the gut of high-fat diet-fed ApoE-/- mice. Moreover, *Bifidobacterium* increased count inversely correlated with plaque/lumen area. This implies that tea polyphenols specifically target *Bifidobacterium* to decrease atherosclerotic plaque [183]. Lastly, tea polyphenols were found to promote short-chain fatty acids production in high-fat diet-fed mice [184], while 2, 3, 5, 4′-Tetrahydroxy-stilbene-2-O-β-D-glucoside (TSG) on the GM–atherosclerotic disease axis, modulated gut microbiome composition (*Proteobacteria, Bacteroidetes, Firmicutes, Tenericutes,* and the species *Helicobacter pylori* and *Akkermasia muciniphila*). Furthermore, TSG remarkably inhibited atherosclerotic plaque formation, improved lipid dysregulation, and suppressed inflammation to combat atherosclerotic in ApoE-/- mice [184]. However, further studies are required to establish whether gut microbiota modulation is the main mechanism by which TSG attenuates atherosclerosis.

## 10. Conclusions

Atherosclerosis has complex baseline mechanisms; therefore, the development of therapeutic strategies for this disease is difficult. Polyphenols represent promising agents in the prevention and treatment of atherosclerosis, as demonstrated by in vitro, animal, preclinical and clinical studies. The scavenging activity of polyphenols towards reactive oxygen species theory is currently the most validated for explaining the beneficial effects of these compounds to-wards various types of lifestyle diseases, including atherosclerosis. Polyphenolics also display indirect antioxidant activity through the activation of Nrf2. The anti-inflammatory properties of phenolic compounds are strongly connected with the modulation of oxidative stress and in maintaining cell redox homeostasis. This group of compounds can decrease the cellular production of pro-inflammatory mediators. Moreover, polyphenols are able to restrict the expression of adhesion molecules, thus impairing the chemotaxis of monocytes within the inflamed tissues. 

However, most polyphenols cannot be absorbed directly by the small intestine. Therefore, their bioavailability and impact on the host mostly depend on their conversion. Gut microbiota plays a crucial role in converting dietary polyphenols into absorbable bioactive substances. 

In addition, these compounds can affect the composition of gut microbiota. Studies have shown that the administration of tea polyphenols regulated gut microbial dysbiosis and also increased the number of beneficial microbes such as *Lactobacillus*, *Akkermansia, Blautia*, *Roseburia* and *Eubacterium*. The consumption of a coffee preparation of green and roasted coffee beans promoted metabolic activity and enhanced the number of *Bifidobacterium* spp. population. Compared to baseline, the daily intake of red wine polyphenols for 4 weeks was found to significantly increase the number of groups of *Enterococcus*, *Prevotella, Bacteroides*, *Bifidobacterium*, *Bacteroides* uniformis, *Eggerthella lenta* and *Blautia coccoides–Eubacterium* rectale. The obtained results suggest possible probiotic benefits of wine. According to a clinical study, hesperidin and naringin can improve intestinal microflora homeostasis by increasing the population of fecal *Bifidobacterium* spp. and *Lactobacillus* spp. The in vitro anti-*Helicobacter pylori* activity of citrus polyphenols such as hesperetin, naringenin, poncirin and diosmetin has also been demonstrated.

An increasing understanding of the field has confirmed that specific GM taxa strains mediate the gut microbiota–atherosclerosis axis. However, more intervention studies in healthy subjects and in subjects at risk of cardiovascular disease or related pathologies are needed. The metabolism and distribution of most polyphenols are still not clarified. In the future perspective, it is necessary to deepen the knowledge of polyphenol bioavailability and bioefficacy. Therefore, further studies are warranted involving human volunteers in large-scale and long-term trials using different approaches (randomised, placebo-controlled studies) to better estimate the actual concentration of polyphenols metabolites in plasma within the context of a regular diet, which includes chronic ingestion of phenolics-rich foods.

Moreover, bioavailability studies, including intestinal flora metabolites, are clearly needed in order to better understand polyphenol bioactivity. In addition, in vitro and in vivo studies using many polyphenols are difficult to standardise due to the chemical instability of this group of compounds, and generally, a good control or reference is needed so the data will bring the research to the right conclusions. It should also be taken into account that polyphenols in food are present simultaneously with other phytochemicals; thus, it is difficult to establish which of the family of compounds present in a given food or food product is responsible for the potential biological effect. Therefore, it is necessary to find reliable biomarkers of the polyphenol subclasses to demonstrate further biological actions or efficacy in atherosclerosis.

## Figures and Tables

**Figure 1 ijms-24-07146-f001:**
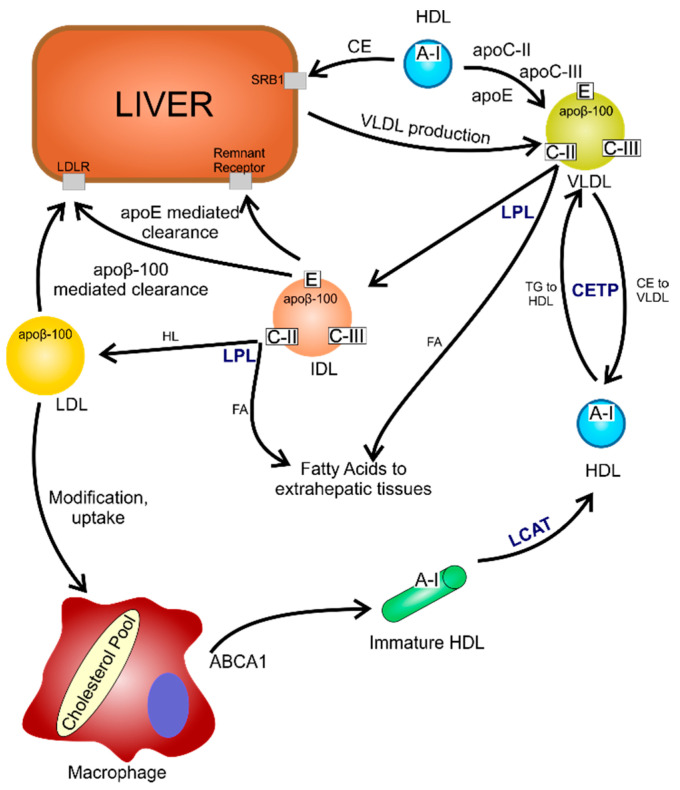
Mechanism of the atherosclerotic process.

**Figure 2 ijms-24-07146-f002:**
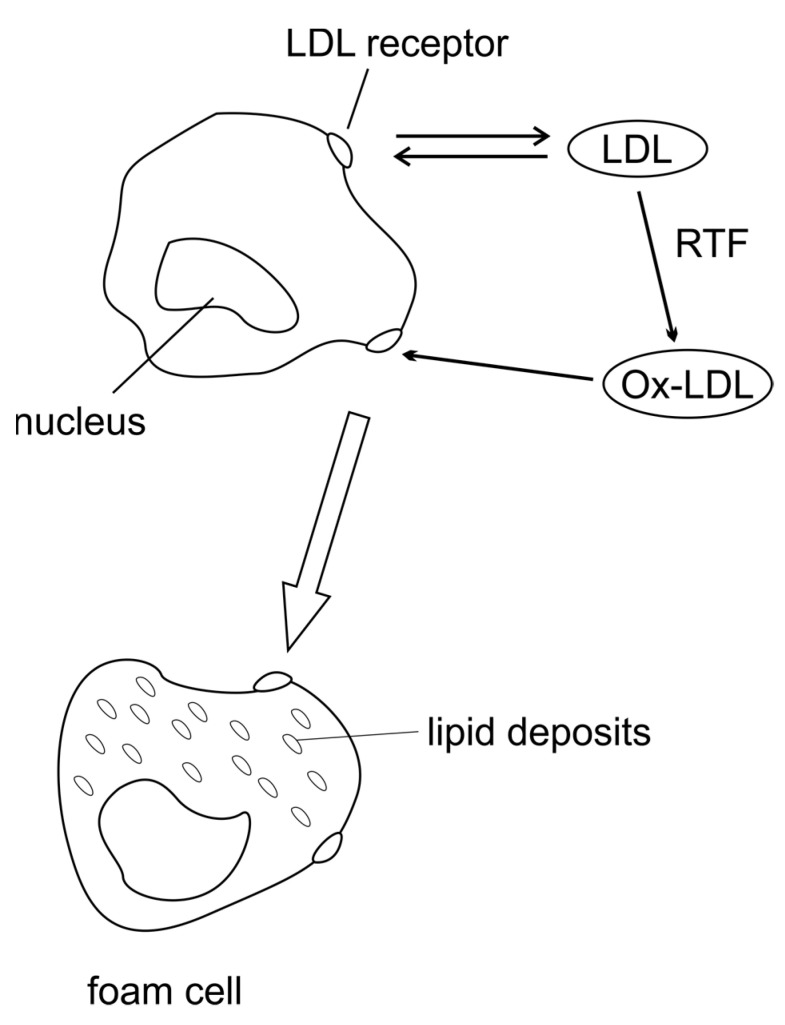
Mechanism of atherosclerosis: ROS modify LDL. Oxidised LDL (Ox-LDL) is formed that interacts with the macrophage scavenger receptor, donating large amounts of lipids to these cells and transforming them into foam cells.

**Figure 3 ijms-24-07146-f003:**
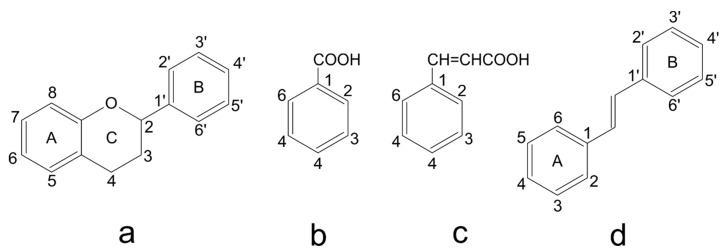
Basic structure of (**a**) flavan, (**b**) benzoic acid, (**c**) cinnamic acid, and (**d**) stilbene.

**Figure 4 ijms-24-07146-f004:**
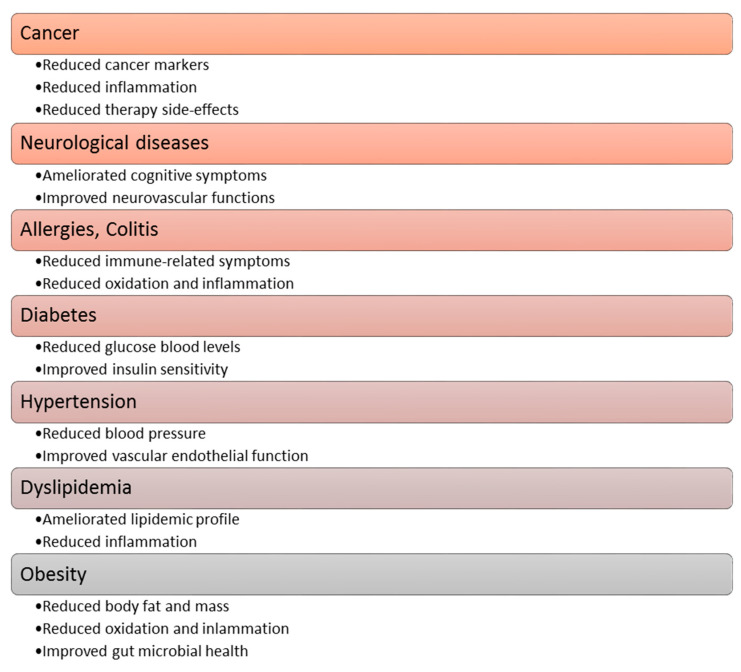
The effect of polyphenolic compounds on selected diseases.

**Figure 5 ijms-24-07146-f005:**
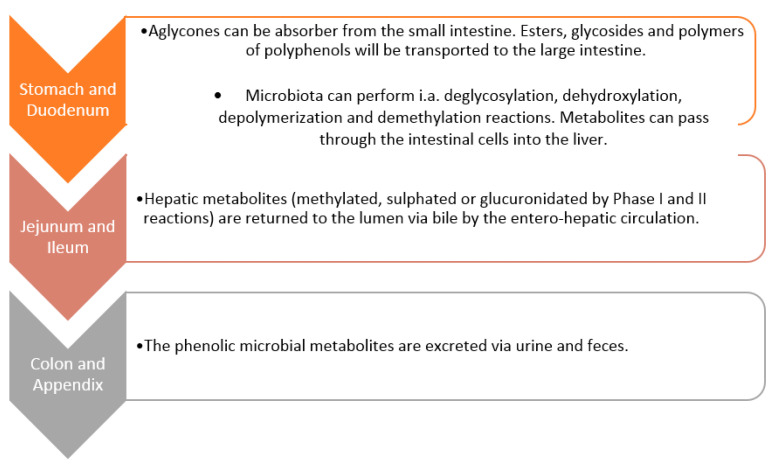
Scheme of microbial metabolism of polyphenolics in the gastrointestinal system.

**Table 1 ijms-24-07146-t001:** Selected important polyphenolic compounds and their sources in the diet.

The Name of the Compound	Structure	Source of Occurrence
Quercetin (flavonoid)	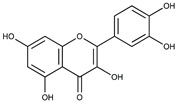	Berries, cherries, grapes, apples, onions, tomatoes, kale and red vine [55].
Rutin (flavonoid glycoside)	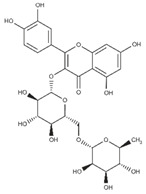	Buckwheat [56] and asparagus [57]
Hesperetin (flavonoid)	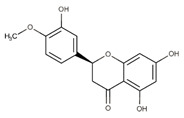	Lime, lemon, sweet orange and tangelo [58]
Naringin (flavonoid)	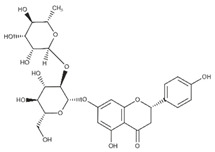	Bergamot and sour orange and grapefruit [58]
Daidzein (Flavonoids)	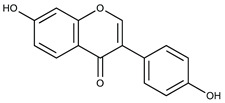	Soy [59]
Cyanidin (flavonoid)	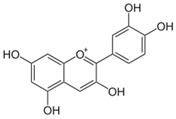	Pomegranate, strawberry, raspberry and bilberry [34]
Protocatechuic Acid (non-flavonoid)	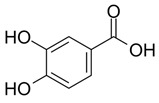	Plums, gooseberries, grape, almonds ordinary, onion, bran and grain brown rice [60]
Ellagic acid (non-flavonoid)	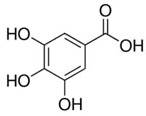	White tea, blackberries and raspberries [61]
Gallic acid (non-flavonoid)	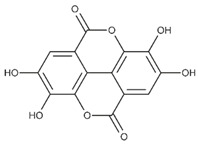	Cranberries, strawberries, blueberries and blackberries [61]
Tannic acid (non-flavonoid)	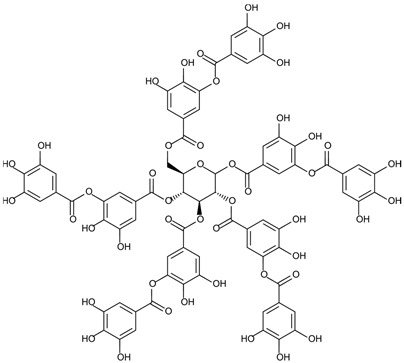	Cranberries, persimmons, almonds, cocoa beans, grape seeds, parsley and peas [61]
Resveratrol (non-flavonoid)	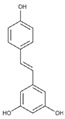	Red grapes, mulberries, pomegranates, blueberries, pistachios and dark chocolate [62]
Curcumin (non-flavonoid)	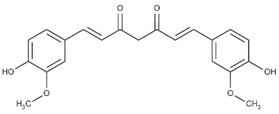	Turmeric and zedoary rhizome [63]

## Data Availability

The data presented in this study are available on request from the corresponding author.

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
