# Peer review of "The Influence of Polyphenols on Atherosclerosis Development"

_ijms, 2023, doi:10.3390/ijms24087146_

Round 1

Reviewer 1 Report

Although the topic is interesting but the article lacks some connectivity between the title and the literature, therefore I have the following suggestions before the further processing of this article.

Abstract:

1.     Abstract needs a revision and modifications because the conclusion of the study is not sufficient as well as try to improve the English grammar.

2.     Recheck keywords

Main body:

1.     Similarity index should no more than 20%, current plagiarism level is very high, try to minimize it.

2.     Introduction should be more wider

3.     Many references are too old, try to refer updated references. I am Suggesting the following references:

1.     Aziz, A., Noreen, S., Khalid, W., Mubarik, F., Niazi, M. K., Koraqi, H., ... & Al-Farga, A. (2022). Extraction of Bioactive Compounds from Different Vegetable Sprouts and Their Potential Role in the Formulation of Functional Foods against Various Disorders: A Literature-Based Review. Molecules27(21), 7320.

2.  Ali, Anwar, Sakhawat Riaz, Aysha Sameen, Nenad Naumovski, Muhammad Waheed Iqbal, Abdur Rehman, Taha Mehany, Xin-An Zeng, and Muhammad Faisal Manzoor. "The disposition of bioactive compounds from fruit waste, their extraction, and analysis using novel technologies: A review." Processes 10, no. 10 (2022): 2014.

4.     Add a little bit information about the cancer statistics in the cancer portion of the study (page number 10) and refer the following suggestion:

1.     Ali, Anwar, Muhammad Faisal Manzoor, Nazir Ahmad, Rana Muhammad Aadil, Hong Qin, Rabia Siddique, Sakhawat Riaz et al. "The burden of cancer, government strategic policies, and challenges in Pakistan: A comprehensive review." Frontiers in Nutrition (2022): 1553.

5.     Source of Polyphenols and effect on the indicators of Atherosclerosis

6.     Add the following reference in the (4. Polyphenolic Compounds as a Considerable Plant Component of the Daily Diet) section:

1.     Ali, Anwar, Safura Kousar, Waseem Khalid, Zahra Maqbool, Afifa Aziz, Muhammad Sajid Arshad, Rana Muhammad Aadil, Monica Trif, Hong Qin, and Muhammad Faisal Manzoor. "Crocin: Functional characteristics, extraction, food applications and efficacy against brain related disorders." Frontiers in Nutrition (2022): 3015.

7.      Polyphenol indicators (occurring compounds) and Atherosclerosis indicator should be more clear (Proper linkage)(Effect)

8.     Diagrams should be more fine if graphical presentation added, will enhance the concept.

9.      Human consumption of polyphenol should be mentioned

10.   How much intake of polyphenols or in which form reduce the risk of Atherosclerosis

11.   Treatment and prevention of Atherosclerosis through polyphenols compounds should be more clear and recommended.

12.  Improve English grammar

13.  Conclusion part is not clear, try to add a future perspective of the study.

Author Response

The authors would like to thank the Reviewer for valuable comments which have helped to improve the quality of the manuscript. We hope that the revisions in the manuscript and accompanying responses will be sufficient to make our manuscript suitable for publication. We have made all the changes suggested in the Reviewer's comments in the text.

Although the topic is interesting but the article lacks some connectivity between the title and the literature, therefore I have the following suggestions before the further processing of this article.

Abstract:

  1. Abstract needs a revision and modifications because the conclusion of the study is not sufficient as well as try to improve the English grammar.

We thank the Reviewer for comments. Now the abstract has been rebuilt to reflect the clou of our review paper. The manuscript has been improved towards grammar and stylistics by native speaker Jack Stanley Dunster from Canada (Language Editor of Current Issues in Pharmacy and Medical Sciences), who has many years of experience in this type of work.

  1. Recheck keywords

Thank you for the suggestion. Keywords  have been checked and corrected

Main body:

  1. Similarity index should no more than 20%, current plagiarism level is very high, try to minimize it.

Thank you for valuable comments We tried to minimize phrases taken too literally from other articles.

  1. Introduction should be more wider

Thank you for valuable comment. The introduction has been extended to the possibilities of pharmacological treatment of atherosclerosis

  1. Many references are too old, try to refer updated references. I am Suggesting the following references:Aziz, A., Noreen, S., Khalid, W., Mubarik, F., Niazi, M. K., Koraqi, H., ... & Al-Farga, A. (2022). Extraction of Bioactive Compounds from Different Vegetable Sprouts and Their Potential Role in the Formulation of Functional Foods against Various Disorders: A Literature-Based Review. Molecules, 27(21), 7320.Ali, Anwar, Sakhawat Riaz, Aysha Sameen, Nenad Naumovski, Muhammad Waheed Iqbal, Abdur Rehman, Taha Mehany, Xin-An Zeng, and Muhammad Faisal Manzoor. "The disposition of bioactive compounds from fruit waste, their extraction, and analysis using novel technologies: A review." Processes10, no. 10 (2022): 2014.

Thank you for the suggestion. References are supplemented with proposed articles, and the oldest items are removed.

  1. Add a little bit information about the cancer statistics in the cancer portion of the study (page number 10) and refer the following suggestion: Ali, Anwar, Muhammad Faisal Manzoor, Nazir Ahmad, Rana Muhammad Aadil, Hong Qin, Rabia Siddique, Sakhawat Riaz et al. "The burden of cancer, government strategic policies, and challenges in Pakistan: A comprehensive review." Frontiers in Nutrition (2022): 1553.

Thank you for the suggestion. Information about the cancer statistics have been added.

  1. Source of Polyphenols and effect on the indicators of Atherosclerosis and 6. Add the following reference in the (4. Polyphenolic Compounds as a Considerable Plant Component of the Daily Diet) section:     Ali, Anwar, Safura Kousar, Waseem Khalid, Zahra Maqbool, Afifa Aziz, Muhammad Sajid Arshad, Rana Muhammad Aadil, Monica Trif, Hong Qin, and Muhammad Faisal Manzoor. "Crocin: Functional characteristics, extraction, food applications and efficacy against brain related disorders." Frontiers in Nutrition (2022): 3015.

Thank you for valuable comments. The reference entitled Crocin: Functional characteristics, extraction, food applications and efficacy against brain related disorders has been added to section 4. Information about human consumption of polyphenols can be found in Table 1 entitled Selected important polyphenolic compounds and their sources in the diet.

  1. Polyphenol indicators (occurring compounds) and Atherosclerosis indicator should be more clear (Proper linkage)(Effect)

Polyphenols play a role in reducing ROS, inflammatory processes such as monocyte adhesion and VSMC proliferation and migration, all of which are key events in atherosclerosis. Now we added and highlighted in the text the potential of several  plant polyphenols and polyphenol reach products and mechanism it's action  based on clinical and pre-clinical studies.

  1. Diagrams should be more fine if graphical presentation added, will enhance the concept.

According to the diagram note, the original Figure 1 has been removed. In its place, two other, better illustrative schemes have been added.

  1. Human consumption of polyphenol should be mentioned

Thank you for the suggestion. Information about human consumption of polyphenol can be found in section 4 (Polyphenolic Compounds as a Considerable Plant Component of the Daily Diet).

  1. How much intake of polyphenols or in which form reduce the risk of Atherosclerosis

Thank you for a valid question. It is impossible to say, in general, what doses of polyphenols reduce the risk of atherosclerosis. It depends on the specific compound, the type of food in which it is served, as well as the condition of the patient - as indicated by the presented clinical studies. Dosages of specific polyphenols that have been shown to be effective in clinical and preclinical studies have been added to the description of individual compounds (section )

  1. Treatment and prevention of Atherosclerosis through polyphenols compounds should be more clear and recommended.

Thank you for the suggestion. We have added a fragment of text highlighting the importance of polyphenols in atherosclerosis (section 8. The impact of polyphenols and their metabolites on the mechanisms and factors causing atherosclerosis). However, it should be emphasized that polyphenols and reach in polyphenols food products  represent only promising agents in the prevention of CVD or as an addion therapy with current treatments. Polyphenols play a role in reducing ROS, inflammatory processes such as monocyte adhesion and VSMC proliferation and migration, all of which are key events in atherosclerosis. This review highlighted the potential of several plant polyphenols and polyphenol reach products, including fruits, vegetables, teas and oils, as anti-CVD therapies based on clinical and pre-clinical mechanism-based studies.

  1. Improve English grammar

We thank the Reviewer for comment. The manuscript has been improved towards grammar and stylistics by native speaker Jack Stanley Dunster from Canada (Language Editor of Current Issues in Pharmacy and Medical Sciences), who has many years of experience in this type of work.

  1. Conclusion part is not clear, try to add a future perspective of the study.

Thank you for the suggestion. Conclusions have been corrected and supplemented about future perspective.

Reviewer 2 Report

A review article summarizes current reports on the effects of polyphenols in food on the development, prevention, and treatment of atherosclerosis, and analyses their anti-inflammatory properties. Overall, I found the paper to be well written and well structured. The references are updated and reflect the scope of the articles. Before the article can be accepted for publication, some improvements need to be made.

2.1. Atherosclerosis as an inflammatory disease: In this section, authors should add a figure describing the whole mechanism of  atherosclerotic process. The same remark for 2.2. The impact of oxidative stress on the development of atherosclerosis.

* The interraction between microbial biofilm inside gut microbiota and polyphenols should be more detailed.

* Extra virgin olive oil is particularly rich in polyphenols which could have a protective effects. This should be futher developped

* Conclusion could be improved. Authors sate that Gut microbiota play a crucial role in converting dietary polyphenols into absorbable bioactive substances. How gut microbiota composition could be changed in the presence of polyphenols and what are the possible human interventions for improvements.

Author Response

The authors would like to thank the Reviewer for valuable comments which have helped to improve the quality of the manuscript. We hope that the revisions in the manuscript and accompanying responses will be sufficient to make our manuscript suitable for publication. We have made all the changes suggested in the Reviewer's comments in the text.

A review article summarizes current reports on the effects of polyphenols in food on the development, prevention, and treatment of atherosclerosis, and analyses their anti-inflammatory properties. Overall, I found the paper to be well written and well structured. The references are updated and reflect the scope of the articles. Before the article can be accepted for publication, some improvements need to be made.

2.1. Atherosclerosis as an inflammatory disease: In this section, authors should add a figure describing the whole mechanism of  atherosclerotic process. The same remark for 2.2. The impact of oxidative stress on the development of atherosclerosis.

The proposed figures have been added.

* The interraction between microbial biofilm inside gut microbiota and polyphenols should be more detailed.

The authors would like to thank the Reviewer for comment. Article have been completed with information about microbial biofilm (section 6. Effect of polyphenols on the composition of gut microbiota).

* Extra virgin olive oil is particularly rich in polyphenols which could have a protective effects. This should be futher developped 

Thank you for the suggestion extra virgin olive oil. Information about have been added to sectio 8.9. Mediterranean diet

* Conclusion could be improved. Authors sate that Gut microbiota play a crucial role in converting dietary polyphenols into absorbable bioactive substances. How gut microbiota composition could be changed in the presence of polyphenols and what are the possible human interventions for improvements.

Thank you for the suggestion. Conclusions have been corrected and supplemented.

Round 2

Reviewer 1 Report

The article is now capable for publication. I recommend its publication because the authors have fulfilled all the necessary comments given by myself.